# Genetic perturbation of cellular homeostasis regulates integrated stress response signaling to control *Drosophila* hematopoiesis

Kishalay Ghosh[1,2,*], Rohit Krishnan Iyer[1,*], Saloni Sood[1,3], Mohamed Sabeelil Islam[1], Jyotsana G. Labad[1] and Rohan Jayant Khadilkar[1,4,‡]

## ABSTRACT

Aging results in a decline in cellular and molecular functions. One of the hallmarks of aging is stem cell exhaustion, which impacts self-renewal and differentiation. We employ the *Drosophila* larval lymph gland (LG) to investigate the impact of genetic perturbation of cellular homeostasis on hematopoiesis. The LG consists of a posterior signaling center (PSC) – a stem cell niche that maintains medullary zone (MZ) prohemocytes, whereas the cortical zone (CZ) consists of differentiated hemocytes. We employed over-activation of Toll or Imd pathway to disrupt cellular homeostasis, whereas we over-expressed Foxo or Atg8 to balance it. Genetic perturbation of cellular homeostasis displays hallmarks of aging. Induction of Toll or Imd pathway locally and systemically leads to a decreased niche size and increased differentiation, whereas Foxo or Atg8 over-expression shows an opposite trend. We showed that the integrated stress response (ISR) pathway is induced upon Toll or Imd over-activation and LGs with ISR perturbation show increased hemocyte differentiation. Genetic epistasis shows that ectopic over-expression of ISR components upon Imd activation can rescue hematopoietic defects. Overall, our study explores how genetic perturbation of cellular homeostasis can impact hematopoiesis. Our research has implications in understanding how abrogation of cellular homeostatic mechanisms may lead to onset of malignancies.

KEY WORDS: Homeostasis, Stem cells, Signaling, Hematopoiesis, *Drosophila*

## INTRODUCTION

Aging is characterized by a progressive decline in cellular and molecular functions that impacts organismal function. Physiological disorders like cancer, diabetes, and cardiovascular and neurodegenerative disorders are an outcome of risk factors closely associated with aging (López-Otín et al., 2023). The hallmarks of aging include genomic instability, loss of proteostasis, mitochondrial dysfunction, stem cell exhaustion, etc., which adversely affect organismal lifespan. Loss of cellular homeostasis could adversely affect stem cells and their differentiation. Stemness refers to the self-renewal ability of stem cells and their capacity for multi-directional differentiation, which is gradually affected upon aging (Yi et al., 2020). Stem cells are influenced by the microenvironment or niche in which they reside, which serves as a spatial organization, providing protection and enabling functional interactions. This microenvironment is rich in extracellular matrix (ECM) and signaling molecules, including growth factors and cytokines, which influence stem cell behaviour, thereby impacting their self-renewal. Aging leads to significant changes in the niche including the accumulation of inflammatory cytokines and a decrease in growth factors, alterations in the niche structure due to changes in ECM composition, perturbation in protein homeostasis, mitochondrial dysfunction, accumulation of reactive oxygen species (ROS), DNA damage, cellular senescence, etc. (Farahzadi et al., 2023). Hematopoietic stem cells (HSCs) are known to be impacted by aging. As HSCs age, they show increased myeloid-biased differentiation at the expense of lymphopoiesis (Rossi et al., 2005). Aging also results in remodeling of the bone marrow (BM) niche, where HSCs relocate within the niche, away from the bone surface (endosteum) and to the central regions (Ho and Méndez-Ferrer, 2020). During HSC aging, there is increased accumulation of pro-inflammatory cytokines, impaired autophagy, adipocyte skewing, accumulation of DNA damage and ROS, inactivation of Drp1 (dynamin related protein 1) mediating mitochondrial fission and enhanced mitochondrial oxidative phosphorylation (OXPHOS), all leading to a decline in HSC stemness (Liu et al., 2022). Here, we use *Drosophila* and its larval hematopoietic organ, the lymph gland (LG), to understand the effect of perturbing cellular homeostasis in different blood cell subsets in *Drosophila* both locally and systemically on overall hematopoiesis.

The *Drosophila* hematopoietic system serves as an excellent model due to its striking similarities with the mammalian system (Evans et al., 2003). *Drosophila* hematopoiesis occurs in two distinct waves: The first wave takes place during embryogenesis, where hematopoietic progenitors originate from the pro-cephalic head mesoderm (Evans et al., 2003). These progenitors differentiate into various hemocyte types, primarily plasmatocytes and crystal cells. Plasmatocytes, which account for over 90% of hemocytes, are functionally similar to macrophages, engaging in phagocytosis and immune responses. Crystal cells (5% of hemocytes) are involved in wound healing, the synthesis of melanin through prophenoloxidase (PPO) and are recently reported to be oxygen carriers (Shin et al., 2024). The second wave occurs during larval development, primarily within a specialized organ known as the LG (Lanot et al., 2001; Jung et al., 2005). The LG is an active hub where progenitor cells maintain their population and upon receiving required cues differentiate into mature hemocytes. There is a third blood cell type in *Drosophila* named lamellocytes that are

[1]Stem Cell and Tissue Homeostasis laboratory, Advanced Centre for Treatment, Research and Education in Cancer (ACTREC), Tata Memorial Centre, Kharghar, Navi Mumbai, Maharashtra 410210, India. [2]ICM - Paris Brain Institute, Hôpital Pitié, 47 Bd de l'Hôpital, 75013 Paris, France. [3]Department of Pharmaceutical Sciences, University of Arkansas for Medical Sciences, Little Rock, AR 72205, USA. [4]Homi Bhabha National Institute, Training School Complex, Anushakti Nagar, Mumbai, 400085, India.
*Indicates equal first author

‡Author for correspondence (rkhadilkar@actrec.gov.in)

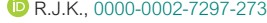 R.J.K., 0000-0002-7297-2736

produced under acute stress conditions like wasp infestation (Rizki and Rizki, 1979; Franc et al., 1996). The LG is organized into distinct zones: the posterior signaling center (PSC), a niche regulating both self-renewal and differentiation of prohemocytes; the medullary zone (MZ), which houses quiescent prohemocytes; an intermediate zone (IZ), which contains hemocytes transitioning towards mature differentiated cells; and a cortical zone (CZ) for mature differentiated hemocytes, namely plasmatocytes, crystal cells and lamellocytes (Jung et al., 2005; Kharrat et al., 2022). Recent single-cell RNA sequencing studies demonstrated a considerable transcriptomic and functional heterogeneity; for example we now know that the progenitors in MZ could be either core or distal hematopoietic progenitors (Fu et al., 2020; Cattenoz et al., 2020; Tattikota et al., 2020; Cho et al., 2020; Girard et al., 2021; Kharrat et al., 2022). Multiple signaling pathways maintain the PSC niche and regulate MZ resident prohemocyte differentiation. MZ Prohemocytes are maintained by Hedgehog- and Serrate-mediated Notch signaling from the PSC (Lebestky et al., 2003; Mandal et al., 2007; Baldeosingh et al., 2018). Signaling pathways like wingless (Sinenko et al., 2009; Goins et al., 2024), Dpp (Pennetier et al., 2012; Dey et al., 2016), and JAK/STAT (Minakhina et al., 2011; Sinha et al., 2013; Rodrigues et al., 2021) also regulate prohemocytes in the MZ. Such tight regulation maintains a balance between stem cell quiescence and activation, ensuring a steady supply of progenitors for differentiation into various hemocyte types (Duvic et al., 2002; Benmimoun et al., 2012; Pennetier et al., 2012; Banerjee et al., 2019; Kharrat et al., 2022). Similarly, a complex interplay of different LG-intrinsic and -extrinsic signaling pathways determine the extent of differentiation (Koranteng et al., 2022). For example, Pvr (PDGF/ VEGF), EGFR promotes plasmatocyte differentiation while Notch and Hippo signaling are crucial for crystal cell differentiation (Milton et al., 2014; Dey et al., 2016; Khadilkar and Tanentzapf, 2019; Zhang et al., 2023). Under stressful conditions like increased ROS levels, injury, infection or wasp infestation, signals like JNK, Toll, and EGFR pathway are activated to promote lamellocyte differentiation, a hemocyte type found only in such situations (Meister and Ferrandon, 2011; Anderl et al., 2016; Louradour et al., 2017). The complexity and heterogeneity of cell subsets in the LG coupled with the genetic tools available for gene manipulation in the *Drosophila* hematopoietic system provide a scorable system to assess the impact of genetically modulating cellular aging on blood cell homeostasis.

In order to understand if age-associated decline in cellular functions impacts organ physiology or not, we have used LG, the *Drosophila* hematopoietic organ. Using localized and systemic genetic perturbation of molecular regulators of cellular homeostasis, we set out to understand the effect of modulating cellular homeostasis. We employ chronic and persistent activation of inflammation, known to disrupt cellular homeostasis (Ruland, 2011; Balistreri et al., 2013). Chronic activation of inflammation negatively influences overall lifespan and alters stem cell function by promoting a chronic hyperactive immune state with increased expression of anti-microbial peptides and pro-inflammatory cytokines, leading to cellular senescence and impaired stem cell regeneration in *Drosophila* (Garschall and Flatt, 2018; Zhang et al., 2021). We activate Toll and Imd in *Drosophila* to disrupt cellular homeostasis. Pirk [peptidoglycan recognition protein (PGRP)-interacting receptor kinase] acts as a negative feedback regulator of Imd (immune deficiency) pathway, which is a component of NF-κB signaling and involved in eliciting immune response against Gram-negative bacteria in the form of anti-microbial peptides (AMPs) like diptericin, drosocin, etc. (Fabian et al., 2021). Pirk interacts with Imd directly and the cytoplasmic tail of PGRP-lysine type (LC) and forms a complex that inactivates them and prevents nuclear

localization of relish, a NF-κB transcription factor, thereby aiding in eliciting a controlled Imd immune response against pathogens (Li et al., 2020). Knockdown of *pirk* leads to upregulation of the Imd pathway (Sciambra and Chtarbanova, 2021) and in staging a chronic and persistent inflammatory state that systemically accelerates aging and reduces lifespan (Giannakou and Partridge, 2007) by inducing neurodegenerative conditions and metabolic alterations (Yamashita et al., 2021), including altered lipid metabolism due to depleting glycogen reserves in fat body, and increased circulation of sugar (Davoodi et al., 2019). Apart from Imd activation, we target the Toll pathway, whose heightened activation leads to elevated AMP expression and decreased immune efficiency (immunosenescence), corresponding to reduced lifespan (Kaur et al., 2020). Toll10B is a gain-of-function mutation of Toll receptor that constitutively activates the Toll pathway (Artero et al., 2003). In order to maintain or reinstate cellular homeostasis, we employ the over-expression of forkhead box O (Foxo) or Atg8, which have been shown to regulate anti-aging (Kenyon, 2010; Bai et al., 2013; Li et al., 2021). Foxo are proteins under the forkhead family of transcription factors and are characterized by a conserved DNA-binding domain known as forkhead box (FOX). They aid in lifespan extension by decreasing insulin-like growth factor (IGF)-like signaling (IIS) – inducing a state similar to caloric restriction (Broughton et al., 2005) – enhancing autophagy (Morselli et al., 2010), upregulating stress-responsive genes and downregulating genes of oxidative phosphorylation to tackle oxidative and metabolic stress (Hwangbo et al., 2004). Foxo has been shown to function as an integrator of cellular homeostasis (Salih and Brunet, 2008; Eijkelenboom and Burgering, 2013). dFoxO over-expression in *Drosophila* fat body and gut under specific dietary conditions extended longevity (Broughton et al., 2005). In *Caenorhabditis elegans*, activation of DAF-16, a FoxO homolog, promoted oxidative stress resistance and enhanced autophagy, thereby extending lifespan (Kenyon, 2010). *Drosophila* Atg8a is an autophagic protein with a crucial role in autophagosome biogenesis and maturation, which is important for degradation and clearance of damaged and accumulated cellular components, thereby maintaining cellular homeostasis and extending lifespan (Yin et al., 2016; Chun and Kim, 2018; Eskelinen, 2019; Perrotta et al., 2020). Also, it binds to transcription factors like sequoia, which regulates autophagy gene expression (Jacomin et al., 2020). Atg8a also interacts with sirtuins, which are upregulated in conditions of nutrient stress and then deacetylate Atg8a to activate autophagy under starvation conditions (Jacomin et al., 2020). Atg8a over-expression in specific tissues like muscles and gut resulted in enhanced tissue integrity and function and systemically led to an overall organismal longevity and improved health span (Bai et al., 2013; Li et al., 2021). Atg8a is also known to promote mitochondrial function by upregulating genes involved in mitochondrial biogenesis and mitophagy (Youle and van der Bliek, 2012).

In this study, we conducted a detailed characterization of how *Drosophila* LG hematopoiesis is impacted upon genetic perturbation of cellular homeostasis, both locally and systematically, and in different cell subsets. We validated these genetic modalities and also used alternate approaches, including chemical or drug-based interventions for modulating cellular homeostasis. Furthermore, we uncovered a signaling mechanism that activates when the cells in the LG sense stress induced by Toll or Imd activation. Our results also show that ectopic integrated stress response (ISR) pathway activation, over and above its existing levels, is capable of restoring hematopoiesis in the LG during stress

induced by chronic Toll or Imd activation. Our results provide important insights into the biology of cellular aging and homeostasis of the stem cell niche microenvironment, which often transforms into a microenvironment similar to a leukemogenic niche that fosters malignancies in such adverse conditions.

## RESULTS

### Hematopoietic progenitor-specific genetic perturbation of regulators of organismal aging disrupts cellular homeostasis

Aging is associated with a progressive decline in cellular and molecular functions that disrupts cellular homeostasis along with increase in mortality (Da Silva and Schumacher, 2021). Here, we set out to understand the effect of modulating the regulators of cellular homeostasis in various cellular subsets of the *Drosophila* larval LG to study the effects on hematopoiesis. Now, previous literature suggests that Imd or Toll pathway upregulation predisposed fly brains to toxic levels of AMPs, increased inflammation and dramatically reduced lifespan (Kounatidis et al., 2017; Khor and Cai, 2020) and over-expression of Atg8 in neurons or Foxo over-expression in the pericerebral fat body led to an increase in lifespan at an organismal level in *Drosophila* (Gelino and Hansen, 2012; Du and Zheng, 2021). We set out to understand if these molecular regulators that impact organismal aging in flies have any role to play in regulating cellular homeostasis. We started by functionally validating the effect of genetic perturbation of cellular homeostasis in the blood progenitors in the LG. We upregulated NF-κB signaling pathways like Imd via *pirk* knockdown or Toll pathway by over-expression of Toll10B, a gain-of-function and constitutively active Toll receptor mutant whose over-expression leads to hyperactivation of Toll pathway (Schmid et al., 2014). *pirk* is a negative regulator of *Drosophila* Imd pathway and RNAi-mediated knockdown of *pirk* causes Imd hyperactivation (Kleino et al., 2008). In order to maintain or reinstate cellular homeostasis, we ectopically over-expressed Atg8, an important autophagy-related protein required for autophagosome biogenesis and maturation and induction of autophagy (Nair et al., 2012) and Foxo, a transcription factor regulating diverse biological processes including aging and metabolism (Du and Zheng, 2021). As a functional readout of cellular homeostasis, we have assessed the typical hallmarks of cellular aging including levels of autophagy, reactive oxygen species (ROS) levels and regulation of proteostasis i.e. protein turnover assessment (López-Otín et al., 2023) upon genetic modulation of cellular aging in the *tepIV*-positive core progenitors of LG.

For estimation of levels of autophagy, we scored for p62, an adaptor protein that mediates interaction between the cargo and Atg8 on autophagosomes for cargo degradation and gets cleared along with the cargo during autophagy (Bjørkøy et al., 2009). We have estimated the ratio of p62 positive puncta per cell upon modulation of cellular homeostasis and our results indicate that there are decreased p62 positive puncta upon over-expression of Atg8 in *tepIV*-positive progenitors (Fig. 1B,D; Fig. S5B-B′) whereas an accumulation of p62 upon Imd activation (Fig. 1C,D; Fig. S5C-C′) as compared to the wild-type control (Fig. 1A,D; Fig. S5A-A′). These results indicate that autophagy in the blood progenitors is impaired upon Toll or Imd activation whereas in Foxo or Atg8 over-expression scenario, there is degradation of p62 moiety along with the autophagic cargo suggesting an efficient and functional autophagy process. ROS levels measured as mean fluorescence intensity levels were not altered upon Atg8 or Foxo over-expression (Fig. 1F,G,I) while Toll over-activation in progenitors exhibited accumulation of ROS (Fig. 1H–I) as compared to wild-type control (Fig. 1E,I). For assessment of regulation of proteostasis, protein turnover was

estimated using Proteostat reagent (Basisty et al., 2018; Kakraba et al., 2023) and we find that *pirk* knockdown or Toll pathway upregulation showed significant protein aggregate accumulation (Fig. 1M-O; Fig. S5G-H′) indicating a net decrease in protein turnover and loss of proteostasis whereas Atg8 or Foxo over-expression displayed significant suppression of protein aggregate formation (Fig. 1K,L,O; Fig. S5E-F′) indicating increase in protein turnover and restoration of protein homeostasis as compared to wild-type control (Fig. 1J,O; Fig. S5D-D′).

### Hematopoietic niche size in the LG is altered upon localized or systemic perturbation of cellular homeostasis

PSC serves as the hematopoietic progenitor-niche or microenvironment that interacts with the hematopoietic progenitors through signaling to maintain a balance between progenitor maintenance and differentiation (Krzemień et al., 2007; Mandal et al., 2007; Pennetier et al., 2012). Here, we investigate the effect of perturbing regulators of cellular homeostasis on the size of PSC niche in LG. Previous studies have shown that downregulation of the Imd pathway by Relish depletion leads to PSC niche hyperplasia and aberrant differentiation of progenitors (Ramesh et al., 2021). We first performed PSC-niche specific induction of Toll or Imd pathway and our results indicate that over-activation of Toll using niche- specific *collierGal4* leads to a decrease in Antennapedia-positive cells (Fig. 2E,a; Fig. S6E) whereas Imd activation has no effect on the niche (Fig. 2D,a; Fig. S6D). Niche-specific over-expression of Atg8 has no significant effect on PSC size (Fig. 2B,b; Fig. S6B) whereas Foxo over-expression leads to a decrease in niche size (Fig. 2C,b; Fig. S6C) as compared to the wild type (Fig. 2A,a,b; Fig. S6A). We then assessed if there is any cell non-autonomous effect of progenitor-specific modulation using *domeGal4* on the niche size. Similar to the cell-autonomous effect of Toll activation, progenitor-specific Toll activation resulted in a decrease in niche size whereas Imd activation did not alter niche size as compared to the wild type (Fig. 2F,I,J,c; Fig. S6I-J). Over-expression of Foxo or Atg8 in the progenitors had no effect on niche size as compared to the wild type (Fig. 2F,G,H,d; Fig. S6F-H). Previous studies have shown that differentiated hemocytes in the LG can reciprocally regulate the blood progenitors (Mondal et al., 2011); however, whether the cortical zone hemocytes regulate niche size is underexplored. We hypothesized that modulating cellular homeostasis in mature differentiated hemocytes could remodel the niche in the organ. In order to test this, we used *hmlΔGal4* that drives expression in cortical zone differentiated hemocytes. *hmlΔGal4* mediated activation of Toll or Imd pathway and expression of Foxo or Atg8 has no effect on the niche size as compared to the wild type (Fig. 2K-O,e,f; Fig. S6K-O). We then performed whole LG-specific genetic modulation using *e33cGal4* and found that both Toll or Imd activation resulted in a smaller niche (Fig. 2S,T,g; Fig. S6S,T), whereas Atg8 over-expression had no effect on the niche (Fig. 2Q,h; Fig. S6Q). Foxo over-expression very similar to its cell autonomous effect on the niche led to a decrease in niche size upon LG-specific over-expression (Fig. 2R,h; Fig. S6R) as compared to wild-type control (Fig. 2P,g,h; Fig. S6P). Since LG cells can respond to systemic changes as documented earlier (Shim et al., 2012; Cho et al., 2018; Goyal et al., 2021; Koranteng et al., 2022), we checked if fat body specific modulation of cellular homeostasis that is known to also impact overall lifespan of flies can alter the niche in the LG. We over-activated Toll or Imd pathway in the fat body using *pplGal4* and it showed no alteration in LG niche size (Fig. 2X,Y,I; Fig. S6X,Y) whereas Atg8 over-expression again had no effect on niche size (Fig. 2V,j; Fig. S6V) and Foxo led to a decrease in niche size

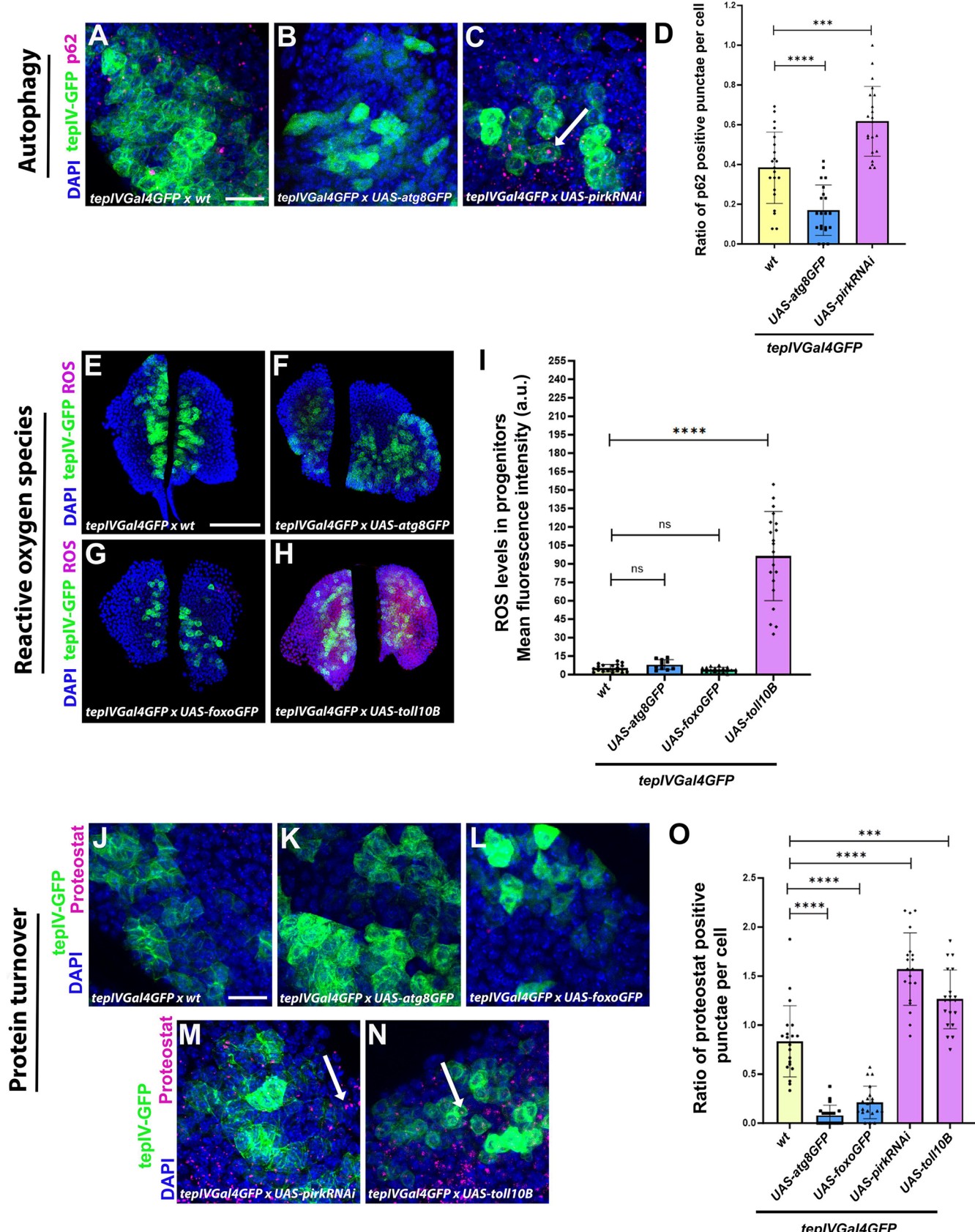

**Fig. 1.** See next page for legend.

**Fig. 1. Validation of genetic perturbation of cellular homeostasis in hematopoietic progenitors of the LG.** Estimation of autophagy levels by evaluating the ratio of p62 positive punctae per cell marked by p62 antibody (magenta) upon core progenitor-specific (using *tepIVGal4*) knockdown of *pirk* or over-expression of *atg8* as compared to wild-type control (A-D). Mean fluorescence intensity of ROS levels in the core progenitors marked by Cell-ROX deep red reagent (red) upon *tepIVGal4* mediated expression of *UAS-atg8GFP, UAS-foxoGFP* or *UAS- toll10B* as compared to wild-type control (E-I). Protein turnover by estimating the ratio of proteostat-positive punctae per cell (magenta) upon *tepIVGal4* mediated expression of *UAS-atg8GFP, UAS-foxoGFP, UAS-pirkRNAi* or *UAS-toll10B* as compared to wild-type control (J-O). Graphical representation of the ratio of p62 positive punctae per cell (D) or mean fluorescence intensity of ROS levels in the *tepIV*-positive core progenitors (I) or ratio of proteostat-positive punctae per cell (O) upon expression of *UAS-atg8GFP, UAS-foxoGFP, UAS-pirkRNAi* or *UAS-toll10B* in the progenitors as compared to wild-type control. For autophagic flux estimation by p62 staining, *tepIVGal4 × UAS-pirkRNAi* (*N*=11, *n*=21) and *tepIVGal4 × UAS-atg8GFP* (*N*=11, *n*=21) were analyzed as compared to wild-type control, *tepIVGal4 × wt* (*N*=11, *n*=21). For estimation of ROS levels by Cell-ROX deep red staining, *tepIVGal4 × UAS-atg8GFP* (*N*=5, *n*=10), *tepIVGal4 × UAS-foxoGFP* (*N*=9, *n*=18) and *tepIVGal4 × UAS-toll10B* (*N*=10, *n*=20) were analyzed as compared to wild-type control, *tepIVGal4 × wt* (*N*=10, *n*=20). For estimation of protein turnover by proteostat detection reagent staining, *tepIVGal4 × UAS-atg8GFP* (*N*=10, *n*=20), *tepIVGal4 × UAS-foxoGFP* (*N*=10, *n*=20), *tepIVGal4 × UAS-pirkRNAi* (*N*=10, *n*=20) and *tepIVGal4 × UAS-toll10B* (*N*=10, *n*=20) were analyzed as compared to *tepIVGal4 × wt* (*N*=10, *n*=20) as wild-type control. *N*, number of larvae; *n*, number of individual primary LG lobes analyzed per genotype. Individual data points in the graphs represent individual primary lobes of the LG. GFP (green) is driven by *tepIVGal4* (A-C,E-H,J-N). Nuclei, DAPI (blue). Values are mean±s.d. and asterisks denote statistically significant differences (ns, not significant, ***$P<0.001$, ****$P<0.0001$). Student's *t*-test with Welch's correction was performed. Scale bars: 30 μm (A-C,J-N), 50 μm (E-H).

(Fig. 2W,j; Fig. S6W) as compared to wild type (Fig. 2U,i,j; Fig. S6U) indicating that perturbation of systemic insulin signaling exerts a long range effect on the LG niche size. Interestingly in the case of Foxo, a downstream target of insulin pathway (Puig and Tjian, 2005) that exerts feedback regulation of the pathway attenuating it (Ni et al., 2007) we find that cell-autonomous, whole LG-specific as well as systemic Foxo over-expression leads to a decrease in niche size demonstrating a negative impact on the insulin signaling that maintains the niche size (Benmimoun et al., 2012). Overall, our results indicate that induction of Toll or Imd pathway decreases the niche size whereas over-expression of Atg8 does not alter the niche size in both localized or systemic manner.

## Modulation of molecular regulators of cellular homeostasis in the PSC niche affects hemocyte differentiation in the LG

Since modulating cellular homeostasis in a localized and systemic manner affected the PSC niche size, we wanted to determine if PSC-specific genetic modulation can impact blood cell differentiation in the LG. PSC-specific modulation was mediated by *collierGal4*. Niche specific activation of Toll or Imd pathway resulted in increased plasmatocyte differentiation (Fig. S1D,E,P) whereas over-expression of Atg8 did not affect plasmatocyte differentiation (Fig. S1B and Q). On the other hand, over-expression of Foxo in the niche that can attenuate insulin signaling increased plasmatocyte differentiation in the LG (Fig. S1C,Q) as compared to the wild-type control (Fig. S1A, P,Q). Imd activation in the niche triggered crystal cell differentiation (Fig. S1I,R), whereas Toll pathway activation resulted in a decrease in crystal cells (Fig. S1J,R). Foxo over-expression in the PSC resulted in a decrease in crystal cell differentiation (Fig. S1H,S), whereas Atg8 over-expression had no significant effect (Fig. S1G,S) as compared to the control (Fig. S1F,R,S). Both Toll or Imd activation resulted in

LGs positive for lamellocytes (Fig. S1N-O′,T) whereas Foxo over-expression in the niche led to a modest 15% LGs being positive for lamellocytes (Fig. S1M-M′,T) and Atg8 over-expression caused no induction of lamellocyte differentiation (Fig. S1L-L′,T) as compared to the wild type (Fig. S1K-K′,T). Previous studies have shown that wasp parasitism increases ROS levels activating Toll and EGFR pathway in the PSC, which promotes progenitor differentiation into plasmatocytes and lamellocytes (Louradour et al., 2017) and trans-differentiation of circulating plasmatocytes into lamellocytes directly on the parasitic wasp eggs via increased Spitz secretion and EGFR signaling (Meister and Ferrandon, 2011; Anderl et al., 2016). The above literature supports our observation of elevated plasmatocyte and lamellocyte differentiation upon Toll pathway activation in the PSC niche.

## Hematopoietic progenitor- specific modulation of cellular homeostasis regulates overall LG hematopoiesis in *Drosophila*

Hematopoietic progenitors in the MZ are important cellular subsets that form the basis of tissue homeostasis in the LG. Hematopoietic progenitors not only respond to localized signals like Wg (Sinenko et al., 2009; Goins et al., 2024), Hh (Mandal et al., 2007; Dey et al., 2016; Baldeosingh et al., 2018), DPP (Pennetier et al., 2012; Dey et al., 2016), JAK-STAT (Minakhina et al., 2011; Sinha et al., 2013; Rodrigues et al., 2021), etc., for their maintenance but are also capable of sensing systemic signals (Benmimoun et al., 2012; Shim et al., 2012; Shim et al., 2013; Cho et al., 2018; Koranteng et al., 2022). Since our results indicated that the hematopoietic niche is impacted upon genetic modulation of cellular homeostasis, we set out to ascertain the cell-autonomous impact on the progenitors itself. Here, we focused on modulating cellular homeostasis in *Domeless*-positive distal progenitor population and *Chiz*-positive intermediate progenitors in the LG. Induction of Toll or Imd activation mediated by *domeGal4* led to an increase in both plasmatocyte and crystal cell differentiation (Fig. 3D-E′,K-L′ F,M) whereas over-expression of Foxo or Atg8 led to no significant effect on plasmatocyte differentiation whereas crystal cell differentiation was suppressed in the case of Foxo over-expression (Fig. 3B-C′,I-J′,G,N) using *domeGal4* as compared to the wild type (Fig. 3A-A′,H-H′,F,G,M, N). Also, Toll activation using *domeGal4* induced lamellocytes consistently in all LGs (100%) (Fig. 3S-S',T) analyzed while Imd activation resulted in 20% LGs with lamellocytes (Fig. 3R-R′,T) whereas Atg8 over-expression did not give any LGs with lamellocytes (Fig. 3P-P′,T) and Foxo over-expression resulted in 10% LGs (Fig. 3Q-Q′,T) with lamellocytes as compared to the control (Fig. 3O-O′,T). We then induced Toll or Imd pathway in the *Chiz*-positive intermediate progenitor population and observed an increase in plasmatocyte differentiation (Fig. S2D,E,P), whereas crystal cells showed an increase upon Imd activation and a suppression upon Toll pathway activation (Fig. S2I-J′,Q) as compared to the control (Fig. S2A,F-F′,P-Q). Atg8 or Foxo over-expression had no effect on both plasmatocyte or crystal cell differentiation (Fig. S2B,C,G-H′,P,Q) as compared to the control (Fig. S2A,F-F′,P-Q). In terms of lamellocyte differentiation, induction of Toll or Imd pathway led to lamellocyte differentiation in the LGs (Fig. S2N-O′,R) whereas Atg8 or Foxo over-expression had no induction of lamellocytes in the LGs (Fig. S2L-M′,R) as compared to their wild-type control (Fig. S2K-K′,R).

## Systemic perturbation of regulators of cellular homeostasis has an impact on LG hematopoiesis

Organismal aging could potentially impact multiple stem cell systems resulting in deregulated homeostasis. There is active

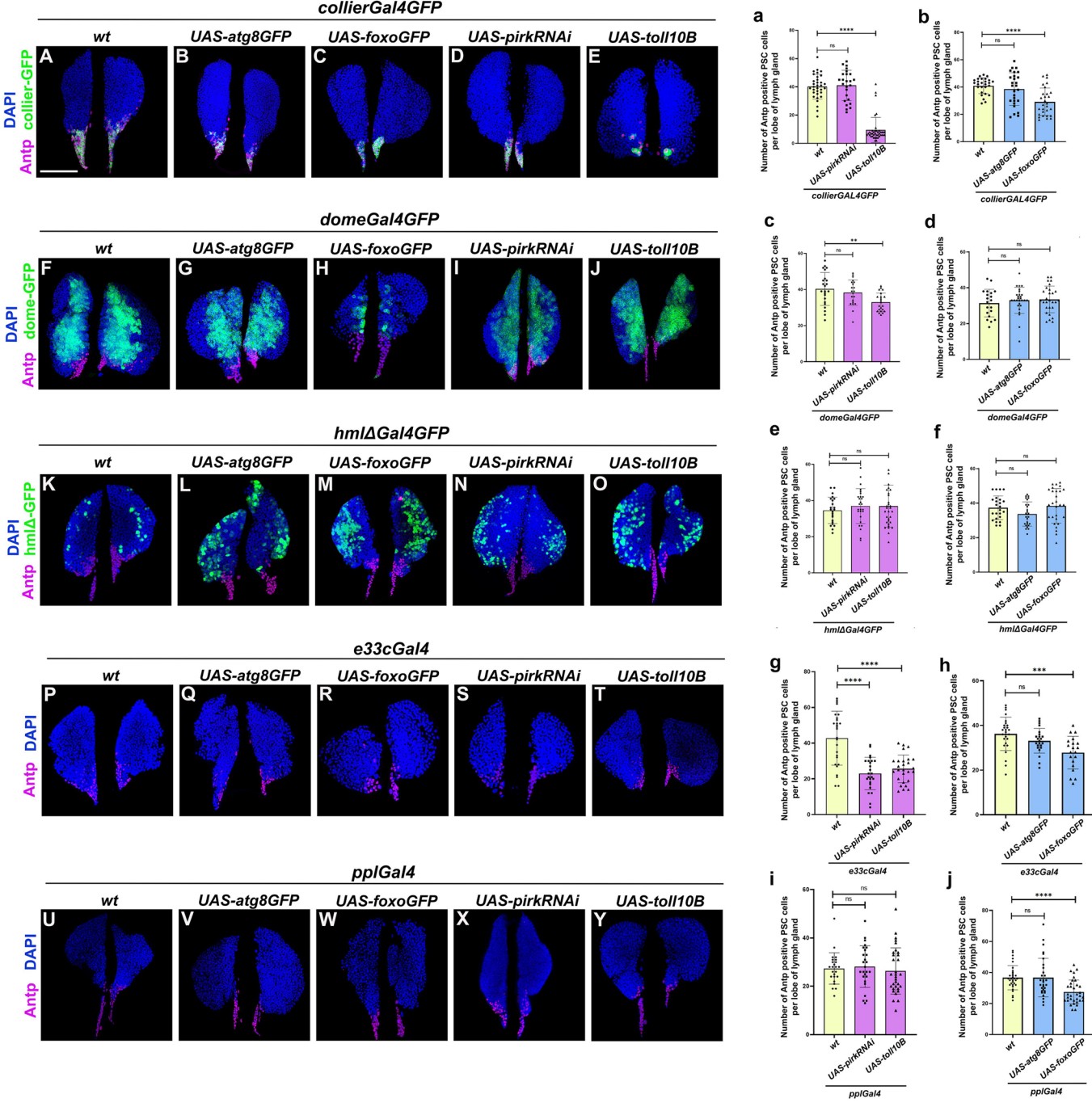

**Fig. 2. Localized perturbation in LG or systemic modulation of cellular homeostasis affects PSC niche size in *Drosophila*.** PSC niche cell population (antennapedia, magenta) and niche cell numbers upon either PSC-specific (using *collierGal4*) or hematopoietic progenitor-specific (using *domeGal4*) or CZ differentiated hemocytes-specific (using *hmlΔGal4*) or whole lymph-specific (using *e33cGal4*) or systemic fat body-specific (using *pplGal4*) expression of *UAS-atg8GFP* (B-V, b-j), *UAS-foxoGFP* (C-W, b-j), *UAS-pirkRNAi* (D-X, a-i) or *UAS-toll10B* (E-Y, a-i) as compared to respective wild-type control (A-U, a-j). For PSC niche-specific genetic modulation, *collierGal4 × UAS-atg8GFP* (*N* =14, *n*=27), *collierGal4 × UAS-foxoGFP* (*N*=15, *n*=29), *collierGal4 × UAS-pirkRNAi* (*N*=13, *n*=25) and *collierGal4 × UAS-toll10B* (*N*= 20, *n*=40) were analyzed as compared to *collierGal4 × wt* (*N*=17, *n*=33). For hematopoietic progenitor-specific genetic modulation, *domeGal4 × UAS-atg8GFP* (*N*=14, *n*=28), *domeGal4 × UAS-foxoGFP* (*N*=14, *n*=28), *domeGal4 × UAS-pirkRNAi* (*N*=11, *n*=21) and *domeGal4 × UAS-toll10B* (*N*=11, *n*=22) were analyzed as compared to *domeGal4 × wt* (*N* =12, *n*=23). For differentiated hemocyte-specific genetic modulation, *hmlΔGal4 × UAS-atg8GFP* (*N*=15, *n*=30), *hmlΔGal4 × UAS-foxoGFP* (*N*=15, *n*=29), *hmlΔGal4 × UAS-pirkRNAi* (*N*=14, *n*=28) and *hmlΔGal4 × UAS-toll10B* (*N*=16, *n*=32) were analyzed as compared to *hmlΔGal4 × wt* (*N*=14, *n*=28). For whole LG-specific genetic modulation, *e33cGal4 × UAS-atg8GFP* (*N*=12, *n*=23), *e33cGal4 × UAS-foxoGFP* (*N*= 10, *n*=20), *e33cGal4 × UAS-pirkRNAi* (*N*=12, *n*=23) and *e33cGal4 × UAS-toll10B* (*N*=14, *n*=27) were analyzed as compared to *e33cGal4 × wt* (*N*=14, *n*=27). For fat body-specific genetic modulation, *pplGal4 × UAS-atg8GFP* (*N*=15, *n*=309), *pplGal4 × UAS- foxoGFP* (*N*=17, *n*=33), *pplGal4 × UAS-pirkRNAi* (*N*=13, *n*=26) and *pplGal4 × UAS- toll10B* (*N*=16, *n*=32) were analyzed as compared to *pplGal4 × wt* (*N*=16, *n*=32). Individual data points in the graphs represent individual primary lobes of the LG. Nuclei, DAPI (blue). GFP is either driven by *collierGal4* (A-E), *domeGal4* (F-J) or *hmlΔGal4* (K-O). Values are mean ±s.d. and asterisks denote statistically significant differences (ns, not significant, **$P<0.01$, ***$P< 0.001$, ****$P<0.0001$). Student's *t*-test with Welch's correction was performed. Scale bars: 50 µm (A-Y).

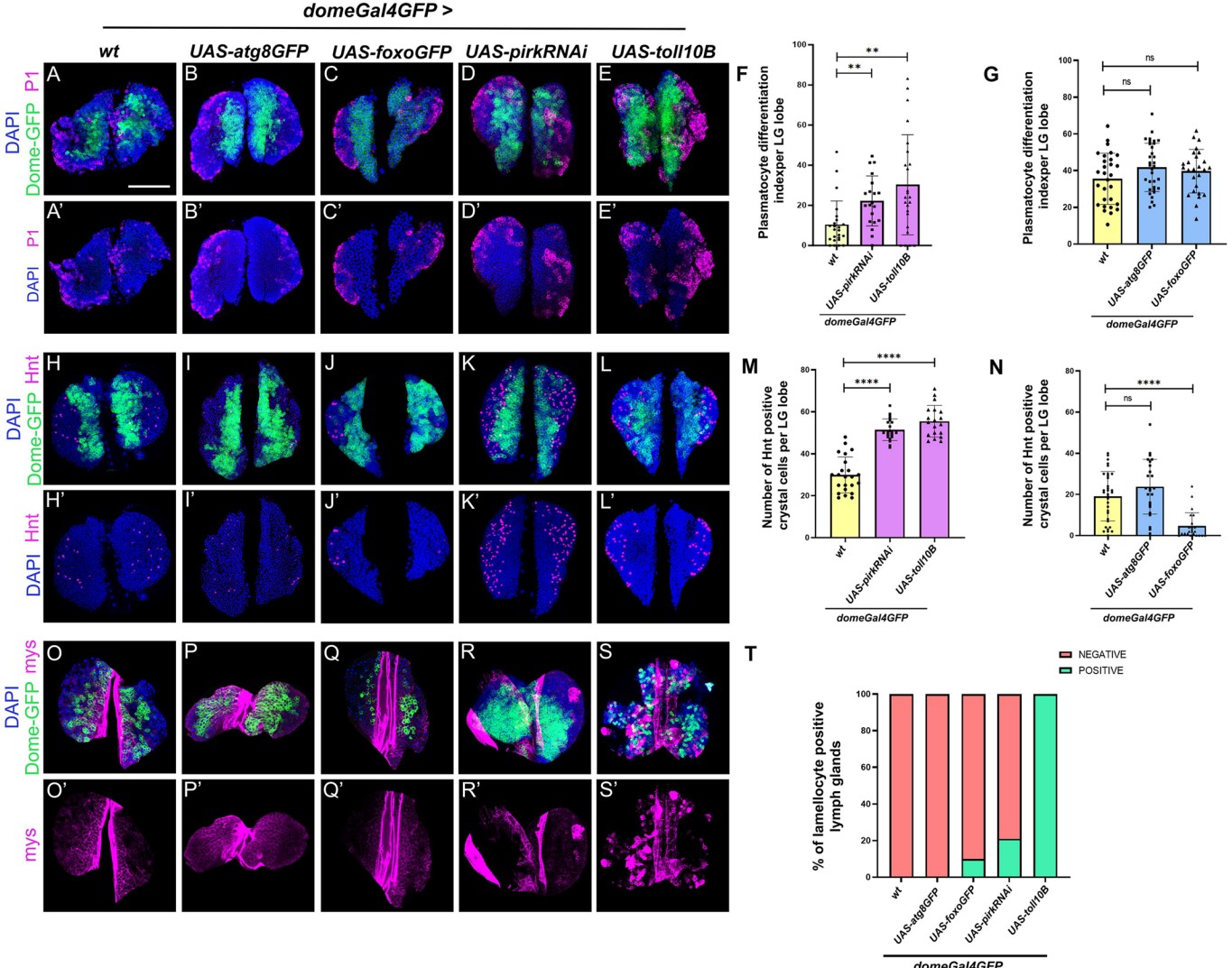

**Fig. 3. Localized distal progenitor subset-specific modulation of cellular homeostasis regulates LG hematopoiesis.** Plasmatocyte differentiation (*P*1, magenta) or crystal cell differentiation (Hnt, magenta) or lamellocyte differentiation (Mys, magenta) upon progenitor-specific (using *domeGal4*) mediated expression of *UAS-atg8GFP* (B,I,P), *UAS-foxoGFP* (C,J,Q), *UAS-pirkRNAi* (D,K,R) or *UAS-toll10B* (E,L,S) as compared to wild-type control (A,H,O). Graphical representation of plasmatocyte differentiation index or number of crystal cells or percentage of lamellocytes-positive LGs upon *domeGal4*-mediated expression of *UAS-pirkRNAi* or *UAS-toll10B* (F,M,T) or *UAS-atg8GFP* or *UAS-foxoGFP* (G,N,T) as compared to wild-type control. For plasmatocyte differentiation: *domeGal4* × *UAS- pirkRNAi* (*N*=10, *n*=20) and *domeGal4* × *UAS-toll10B* (*N*=12, *n*=24) as compared to *domeGal4* × *wt* (*N*=12, *n*=24) and *domeGal4* × *UAS-atg8GFP* (*N*=17, *n*=33), *domeGal4* × *UAS-foxoGFP* (*N*=13, *n*=26) as compared to *domeGal4* × *wt* (*N*=14, *n*=29). For crystal cell numbers: *domeGal4* × *UAS-pirkRNAi* (*N*=11, *n*=21) and *domeGal4* × *UAS-toll10B* (*N*=10, *n*=19) as compared to *domeGal4* × *wt* (*N*=12, *n*=23) and *domeGal4* × *UAS-atg8GFP* (*N*=12, *n*=24), *domeGal4* × *UAS-foxoGFP* (*N*=12, *n*=23) as compared to *domeGal4* × *wt* (*N*=17, *n*=33). *N*, number of larvae; *n*, number of individual primary LG lobes analyzed per genotype. Individual data points in the graphs represent individual primary lobes of the LG. GFP (green) is driven by *domeGal4* (A-S). Nuclei, DAPI (blue). Values are mean±s.d. and asterisks denote statistically significant differences, ns denotes not significant (**P*<0.01, ****P*<0.001, *****P*<0.0001). Student's *t*-test with Welch's correction was performed. Scale bar: 50 µm (A-S').

systemic inter-organ communication and crosstalk between various organs. LG hemocytes have been shown to be responsive to external stimuli from peripheral organs (Benmimoun et al., 2012; Shim et al., 2012; Yang et al., 2015; Cho et al., 2018; Koranteng et al., 2022). Various reports point towards the involvement of fat body and muscles in determining overall organismal lifespan (Demontis and Perrimon, 2010; Bai et al., 2012; Owusu-Ansah et al., 2013; Gáliková and Klepsatel, 2018; Guo et al., 2023). Here, we have used muscles as systemic sites for over-activation of Toll or Imd and over-expression of Foxo or Atg8 and tested the effects of the perturbation on blood cell homeostasis in the LG. We have used *mhcGal4* which expresses in all the muscles including the cardiac muscles in the dorsal vessel. Muscle- specific activation using

*mhcGal4* of both Toll or Imd leads to aberrant differentiation of plasmatocytes (Fig. 4D,E,P), an increase in crystal cells is observed upon Imd activation (Fig. 4I,Q) whereas no effect is seen upon Toll activation (Fig. 4J,Q) as compared to the control (Fig. 4A, F,P,Q). For lamellocytes, both Toll and Imd activation in the muscles triggers lamellocyte production (Fig. 4N-O',R) in the LGs as compared to the control (Fig. 4K-K',R). Interestingly, muscle-specific Foxo over-expression leads to a decrease in both plasmatocyte and crystal cell differentiation and no induction of lamellocytes in the LGs (Fig. 4C,H,M-M',P-R). Atg8 over-expression does not cause any significant effect on the hemocyte differentiation (Fig. 4B,G,L-L',P-R) as compared to the wild-type control (Fig. 4A,F,K-K',P-R).

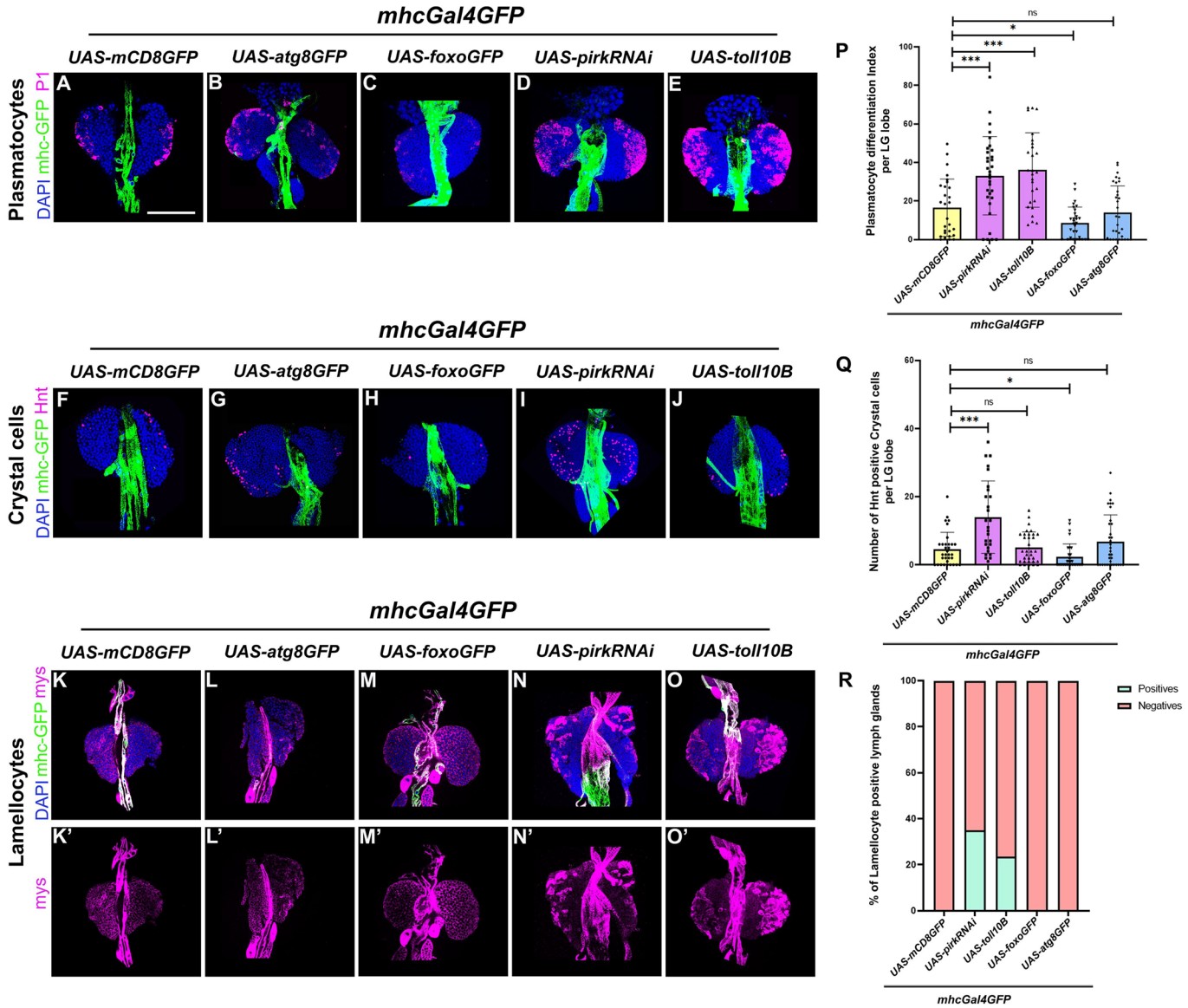

**Fig. 4. Muscle-specific perturbation of molecular regulators of cellular homeostasis affects hemocyte differentiation in the LG.** Plasmatocyte differentiation (P1, magenta) or crystal cell differentiation (Hnt, magenta) or lamellocyte differentiation (Mys, magenta) upon systemic muscle-specific myosin heavy chain (MHC, *mhcGal4*) mediated expression of *UAS-atg8GFP* (B,G,L), *UAS-foxoGFP* (C,H,M), *UAS-pirkRNAi* (D,I,N) or *UAS-toll10B* (E,J,O) as compared to wild-type control (A,F,K). Graphical representation of plasmatocyte differentiation index (P) or number of crystal cells (Q) or percentage of lamellocyte positive LGs (R) upon *mhcGal4GFP* mediated expression of *UAS-atg8GFP, UAS-foxoGFP, UAS- pirkRNAi* or *UAS-toll10B* as compared to wild-type control. For plasmatocyte differentiation, *mhcGal4GFP × UAS-atg8GFP* (*N*=16, *n*=32), *mhcGal4 × UAS-foxoGFP* (*N*=15, *n*=30), *mhcGal4 × UAS-pirkRNAi* (*N*=16, *n*=32) and *mhcGal4 × UAS-toll10B* (*N*=17, *n*=34) were analyzed as compared to *mhcGal4 × UAS-GFP* (*N*=14, *n*=28). For crystal cell numbers, *mhcGal4 × UAS-atg8GFP* (*N*=17, *n*=34), *mhcGal4 × UAS-foxoGFP* (*N*=16, *n*=31), *mhcGal4 × UAS-pirkRNAi* (*N*=15, *n*=30) and *mhcGal4 × UAS-toll10B* (*N*=17, *n*=33) were analyzed as compared to *mhcGal4 × UAS-GFP* (*N*=17, *n*=33). For lamellocyte differentiation, *mhcGal4 × UAS-atg8GFP* (*N*=19), *mhcGal4 × UAS-foxoGFP* (*N*=17), *mhcGal4 × UAS-pirkRNAi* (*N*=20) and *mhcGal4 × UAS-toll10B* (*N*=17) were analyzed as compared to *mhcGal4 × UAS-GFP* (*N*=16). *N*, number of larvae; *n*, number of individual primary LG lobes. Individual data points in the graphs represent individual primary lobes of the LG. GFP (green) is driven by *mhcGal4* (A-O). Nuclei, DAPI (blue). Values are mean±s.d. and asterisks denote statistically significant differences, ns, not significant (\*\**P*<0.01, \*\*\**P*<0.001, \*\*\*\**P*<0.0001). Student's *t*-test with Welch's correction was performed. Scale bar: 50 µm (A-O').

## Chemical-based interventions that perturb cellular homeostasis modulate blood cell differentiation in the LG

In order to supplement our findings on genetic modulation of cellular homeostasis we also used chemical interventions as an alternate approach to modulate homeostasis. Here, we have used Rapamycin and Bortezomib treatment for modulating cellular homeostasis at an organismal level and have investigated its effect on LG hematopoiesis. Previous studies have shown that Bortezomib acts as an inhibitor of the ubiquitin-mediated proteasomal pathway (Chen et al., 2011), thereby resulting in increased protein instability, redox imbalance and accelerated aging (Manola et al., 2019). Additionally, Bortezomib was also shown to induce cellular senescence by stimulating telomere shortening in non-small cell lung cancer (NSCLC) cells (Wang et al., 2022). Bortezomib treatment in wild-type control larvae with green fluorescence marking the *tepIV*-positive core progenitors showed a significant increase in plasmatocyte (Fig. 5A′-A″) and crystal cell differentiation (Fig. 5B′-B″) when compared to its vehicle control (Fig. 5A,A″,B,B″) indicating that there

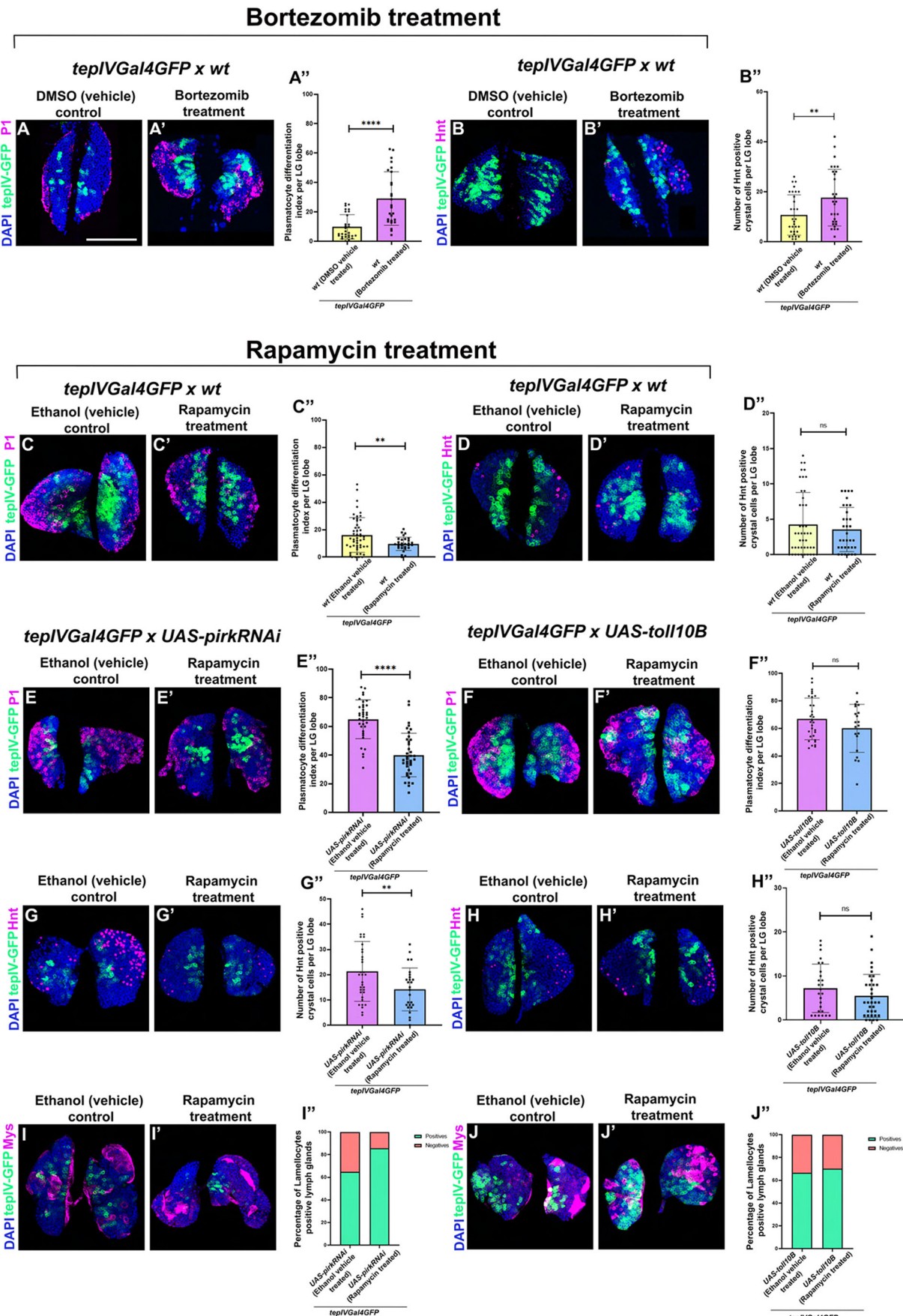

**Fig. 5.** See next page for legend.

**Fig. 5. Chemical modulation of regulators of cellular homeostasis affects hematopoiesis in *Drosophila*.** Plasmatocyte differentiation (P1, magenta) or crystal cell differentiation (Hnt, magenta) in *tepIVGal4 × wt* larvae upon treatment with Bortezomib (A′-B′) as compared with DMSO (vehicle) control (A,B). Graphical representation of plasmatocyte differentiation index (A″) or number of crystal cells (B″) upon Bortezomib treatment as compared with DMSO (vehicle) control. For plasmatocyte differentiation upon bortezomib treatment, *tepIVGal4 × wt* (N=15, n=29) were analyzed as compared to DMSO (vehicle) control (N=13, n=26). Plasmatocyte differentiation (P1, magenta) or crystal cell differentiation (Hnt, magenta) in *tepIVGal4 × wt* (Fig. 6C′,D′) or *tepIVGal4 × UAS-pirkRNAi* (E′, G′) or *tepIVGal4 × UAS-toll10B* (F′,H′) larvae upon treatment with Rapamycin as compared with ethanol (vehicle) control (C,D,E-H). Percentage of Lamellocyte positive LGs in *tepIVGal4 × UAS-pirkRNAi* (I′) or *tepIVGal4 × UAS-toll10B* (J′) upon treatment with Rapamycin as compared with ethanol (vehicle) control (I,J). Graphical representation of plasmatocyte differentiation index or number of crystal cells upon Rapamycin treatment in *tepIVGal4 × wt* (C″-D″) or *tepIVGal4 × UAS-pirkRNAi* (E″,G″) or *tepIVGal4 × UAS-toll10B* (F″,H″) larvae or percentage of lamellocyte positive LGs in *tepIVGal4 × UAS-pirkRNAi* (I″) or *tepIVGal4 × UAS-toll10B* (J″) larvae upon treatment with Rapamycin as compared with ethanol (vehicle) control. For plasmatocyte differentiation upon Rapamycin treatment, *tepIVGal4 × wt* (N=15, n=30), *tepIVGal4 × UAS-pirkRNAi* (N=19, n=38), *tepIVGal4 × UAS-toll10B* (N=10, n=20) were analyzed as compared to respective ethanol (vehicle) controls: *tepIVGal4 × wt* (N=25, n=50), *tepIVGal4 × UAS-pirkRNAi* (N=18, n=36), *tepIVGal4 × UAS-toll10B* (N=17, n=34). For crystal cell differentiation, *tepIVGal4 × wt* (N=20, n=39), *tepIVGal4 × UAS-pirkRNAi* (N=13, n=26), *tepIVGal4 × UAS-toll10B* (N=18, n=35) were analyzed as compared to respective ethanol (vehicle) controls: *tepIVGal4 × wt* (N=24, n=48), *tepIVGal4 × UAS- pirkRNAi* (N=17, n=34), *tepIVGal4 × UAS-toll10B* (N=14, n=28). For Lamellocyte differentiation, *tepIVGal4 × UAS-pirkRNAi* (N=15) and *tepIVGal4 × UAS-toll10B* (N=15) were analyzed as compared to respective ethanol (vehicle) controls: *tepIVGal4 × UAS- pirkRNAi* (N=15) and *tepIVGal4 × UAS-toll10B* (N=15). N, number of larvae; n, number of individual primary LG lobes. Individual data points in the graphs represent individual primary lobes of the LG. GFP (green) is driven by *tepIVGal4* which marks pro-hemocytes (A-J,A′-J′). Nuclei, DAPI (blue). Values are mean±s.d. and asterisks denote statistically significant differences (ns, not significant, **P<0.01, ***P<0.001, ****P<0.0001). Student's t-test with Welch's correction was performed. Scale bar: 50 μm (A-J′).

is perturbation of LG homeostasis upon inhibition of proteostasis. However, rapamycin treatment in wild-type larvae with green fluorescence marking the *tepIV*-positive core progenitors showed a reduction in plasmatocyte differentiation (Fig. 5C′-C″), whereas there was no effect observed on crystal cell differentiation (Fig. 5D′-D″) as compared to its vehicle control (Fig. 5C,C″,D,D″). Previous literature has shown that Rapamycin inhibits mTORC1 (mechanistic target of rapamycin complex 1), which is generally upregulated in amino acid-rich conditions and promotes protein synthesis and inhibits autophagy (Wu et al., 2013). Inhibition of mTORC1 kinase by Rapamycin leads to upregulation of autophagy, increased resistance to starvation and lifespan extension in *Drosophila* (Bjedov et al., 2010). After testing the effect of Rapamycin on wild-type control larvae, we wanted to test if Rapamycin could rescue the blood cell differentiation defects in larvae with Toll or Imd activation in the progenitors. Rapamycin administration to larvae having Imd activation in the *tepIV*-positive core progenitors could rescue plasmatocyte (Fig. 5E′-E″) and crystal cell differentiation significantly (Fig. 5G′-G″) as compared to their control (Fig. 5E,G,E″,G″). However, rapamycin did not display any rescue of differentiation for lamellocytes (Fig. 5I′-I″) when compared to its vehicle control (Fig. 5I,I″). Similarly, administration of rapamycin in larvae with Toll pathway over-activation in *tepIV*-positive core progenitors was unable to rescue the Toll-mediated differentiation of all three hemocyte lineages (Fig. 5F′-J″) as compared to its vehicle control (Fig. 5F-J′,F″-J″), which shows

that Rapamycin can rescue the effects caused by Imd pathway over-activation specifically, which warrants further mechanistic investigation.

## Genetic modulation of cellular homeostasis in hematopoietic progenitors regulates ISR signaling in the LG

The ISR is an important and highly conserved cellular signaling pathway that integrates various signaling pathways that counter various stressors and provides a combined and centralized cellular response that facilitates the organism to adapt to various stresses and promotes maintenance of cellular bioenergetics until the stress is resolved, thereby aiding in survival (Pakos-Zebrucka et al., 2016; Kalinin et al., 2023). Previous studies have shown that as age advances, the ISR pathway is upregulated in the brain of old mice. It was also reported that ISR activation facilitated lifespan extension in *Drosophila* and *C. elegans* (Derisbourg et al., 2022). Interestingly, it has also been found that prolonged ISR activation due to conditions of excessive and persistent stress contributed to cell senescence or programmed cell death (Kalinin et al., 2023). Previous studies have also reported the activation of ISR upon enhanced inflammation and cytokine secretion in mice and ISR activation and phosphorylation of eIF2α was shown to inhibit the translation of cytokine mRNAs in the tissue-resident memory T cells (Trm) of mice (Asada et al., 2024 preprint).

In order to investigate if the stress caused by chronic activation of Toll or Imd that abrogates autophagy, protein turnover and elevates ROS levels results in ISR activation or not, we probed for p-eIF2α levels in the LGs with Toll or Imd activation or upon Atg8 or Foxo over-expression in the hematopoietic progenitors. Our results indicate that the downstream readout of the ISR pathway, i.e. p-eIF2α levels measured by quantitating the mean fluorescence intensity in the *tepIV*-positive core progenitors was significantly increased upon Toll or Imd activation (Fig. 6B-C′,G), whereas over-expression of Atg8 (Fig. 6D-D′,G) or Foxo (Fig. 6E-E′,G) showed no significant difference as compared to the wild-type control (Fig. 6A-A′,G). In addition to this, we also profiled for *gcn2* transcript levels from the hemolymph isolated from larvae where NF-κB signaling pathways or Foxo or Atg8 was induced in all the blood cells using *hmlGal4*, a pan hemocyte *Gal4*. *gcn2* mRNA levels were upregulated upon Imd or Toll pathway activation, while Atg8 or Foxo over-expression showed suppression in the Gcn2 levels as compared to wild-type control (Fig. 6F). This suggests that the upstream ISR activator, Gcn2 and downstream ISR key effector, p-eIF2α are downregulated or inactive in Foxo or Atg8 over-expression genetic background as the cells are already possibly compensating and countering stresses whereas ISR is much needed in an Imd or Toll pathway activated scenario to possibly sense and counter various stresses and promote cell survival.

## Progenitor-specific genetic perturbation of ISR pathway components regulates LG hematopoiesis

Since ISR pathway components and downstream effectors are induced upon Toll or Imd over-activation scenario, we wanted to further test if depletion or over-expression of ISR pathway components in the hematopoietic progenitors has any impact on LG hematopoiesis. Depletion of Gcn2, eIF2α or Atf4 led to an increase in both plasmatocyte and crystal cell differentiation (Fig. 6I,J,K,O,P,Q,Z-Z′) whereas over-expression of *gcn2act* has no effect on plasmatocyte or crystal cell differentiation (Fig. 6L,R,Z-Z′), over-expression of *crcRA* has no effect on plasmatocyte differentiation but results in a decrease in crystal cell differentiation (Fig. 6M,S,Z-Z′) as compared to the wild-type control (Fig. 6H,N,Z-Z′). In terms of

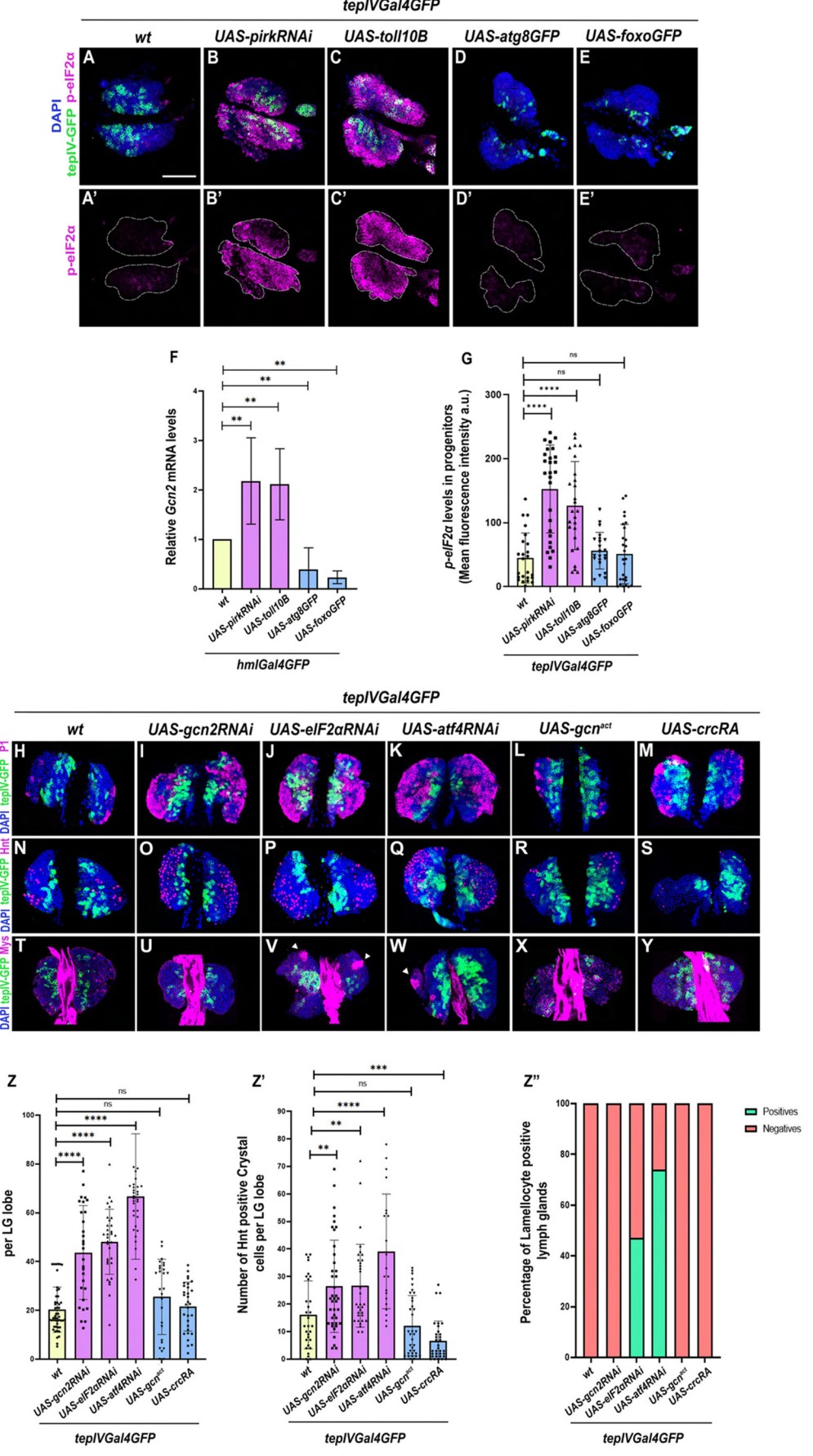

**Fig. 6.** See next page for legend.

**Fig. 6. Hematopoietic progenitor- specific induction of Toll or Imd pathway triggers Integrated Stress Response (ISR) pathway that regulates progenitor differentiation in the LG.** Mean fluorescence intensity of *P*-eIF2α levels in the core progenitors marked by *P*-eIF2α antibody (magenta) upon induction of Toll or Imd pathway using (*UAS-pirkRNAi* or *UAS- toll10B*) or Atg8 or Foxo over-expression (using *UAS-atg8* or *UAS-foxo*) as compared to wild-type control (A-E,A′-E′,G). *P*-eIF2α fluorescence intensity levels upon *tepIVGal4* mediated expression of *UAS-pirkRNAi* or *UAS-toll10B* or expression of *UAS-atg8* or *UAS-foxo* as compared to wild-type control (G). Relative *gcn2* mRNA transcript levels determined by qRT-PCR upon pan-hemocyte-specific (using *hmlGal4*) expression of *UAS-atg8GFP*, *UAS-foxoGFP*, *UAS-pirkRNAi* or *UAS-toll10B* as compared to wild-type control (Fig. 7F). The statistical analysis for qRT-PCR was performed using one-way ANOVA (Dunnett test) for comparison of all test genotypes with wild-type control. Plasmatocyte differentiation (P1, magenta) or crystal cell differentiation (Hnt, magenta) or lamellocyte differentiation (Mys, magenta) upon core progenitor-specific (using *tepIVGal4*) knockdown (*gcn2RNAi, atf4RNAi or eIF2αRNAi*) and over-expression (*gcn2^{act} or crcRA*) of ISR components (I-Y,Z-Z″) as compared to wild-type control (H,N,T,Z-Z″). Graphical representation of plasmatocyte differentiation index (Z) or number of crystal cells (Z′) or percentage of lamellocyte positive LGs (Z″) upon *tepIVGal4* mediated knockdown or over-expression of ISR components as compared to wild-type control. For p-eIF2α levels, *tepIVGal4 × UAS-pirkRNAi* (*N*=14, *n*=27), *tepIVGal4 × UAS-toll10B* (*N*=13, *n*=26), *tepIVGal4 × UAS-atg8GFP* (*N*=11, *n*=22) and *tepIVGal4 × UAS-foxoGFP* (*N*=13, *n*=26) were analyzed as compared to *tepIVGal4 × wt* (*N*=12, *n*=24) as wild-type control. For plasmatocyte differentiation, *tepIVGal4 × UAS-gcn2RNAi* (*N*=17, *n*=34), *tepIVGal4 × UAS-eIF2αRNAi* (*N*=16, *n*=32), *tepIVGal4 × UAS-atf4RNAi* (*N*=16, *n*=32), *tepIVGal4 × UAS-gcn2^{act}* (*N*=15, *n*=30), *tepIVGal4 × UAS-crcRA* (*N*=18, *n*=36) were analyzed as compared to wild-type control, *tepIVGal4 × wt* (*N*=25, *n*=49). For crystal cell differentiation, *tepIVGal4 × UAS-gcn2RNAi* (*N*=20, *n*=40), *tepIVGal4 × UAS-eIF2αRNAi* (*N*=17, *n*=33), *tepIVGal4 × UAS-atf4RNAi* (*N*=13, *n*=25), *tepIVGal4 × UAS-gcn2^{act}* (*N*=18, *n*=35), *tepIVGal4 × UAS-crcRA* (*N*=17, *n*=33) were analyzed as compared to wild-type control, *tepIVGal4 × wt* (*N*=18, *n*=36). For lamellocyte differentiation, *tepIVGal4 × UAS-gcn2RNAi* (*N*=15, *n*=30), *tepIVGal4 × UAS-eIF2αRNAi* (*N*=15, *n*=30), *tepIVGal4 × UAS-atf4RNAi* (*N*=15, *n*=30), *tepIVGal4 × UAS-gcn2^{act}* (*N*=15, *n*=30), *tepIVGal4 × UAS-crcRA* (*N*=15, *n*=30) were analyzed as compared to wild-type control, *tepIVGal4 × wt* (*N*=15, *n*=30). *N*, number of larvae; *n*, number of individual primary LG lobes. Individual data points in the graphs represent individual primary LG lobes. GFP (green) is driven by *tepIVGal4* (A-Y). Nuclei, DAPI (blue). Values are mean±s.d. and asterisks denote statistically significant differences (ns denotes not significant, **P<0.01, ***P<0.001, ****P<0.0001). Student's *t*-test with Welch's correction was performed. Scale bar: 50 µm (A-Y).

lamellocyte differentiation, depletion of eIF2α or Atf4 leads to LGs positive for lamellocytes (Fig. 6V,W,Z″), whereas depletion of Gcn2 or over-expression of *gcn2^{act}* or *crc-RA* (Fig. 6U,X,Y,Z″) does not induce lamellocytes in the LGs as compared to the *wildtype* control (Fig. 6T,Z″).

## Mutants of the ISR pathway components show disruption in blood cell homeostasis

In the previous results, we observed that depletion of ISR pathway components led to aberrant hematopoiesis. Following these observations, we set out to look at the LGs of whole animal mutants of *gcn2* and *crc* (encoding *Drosophila* Atf4). *gcn2^{−/−}* homozygous null mutant and *crc^{1}/+* (cryptocephal) heterozygous hypomorphic allele that have been previously reported were used and their LG phenotypes were analyzed (Vasudevan et al., 2022). *gcn2^{−/−}* homozygous null mutant LGs showed an increase in plasmatocyte (Fig. S3B,C) as well as crystal cell differentiation (Fig. S3E,F) and a presence of lamellocytes in 60% of the LGs (Fig. S3H,I) as compared to *gcn2^{wt}* rescue; *gcn2^{−/−}* as the control (Fig. S3A,D,G,C,F,I). Heterozygotes of *crc^{1}/+* hypomorphic allele displayed a significant

increase in plasmatocyte (Fig. S3K,L) as well as crystal cell differentiation (Fig. S3N,O) along with 80% LGs showing the presence of lamellocytes (Fig. S3Q,R) as compared to wild-type control (Fig. S3J,M,P,L,O,R).

## Hematopoiesis in the LG is sensitive to chemical modulators that regulate the ISR pathway

Since our data on the genetic perturbation of ISR showed an effect on LG hematopoiesis we wanted to validate these findings using known pharmacological modulators of the ISR pathway. We have used the small-molecule ISRIB (ISR Inhibitor) and Histidinol (ISR activator) for our experiments. ISRIB desensitizes eIF2B to the inhibitory effect of p-eIF2α and turns off ISR pathway (Zyryanova et al., 2020) whereas Histidinol, which is an analog of Histidine activates the ISR pathway by mimicking amino acid starvation stress (Taniuchi et al., 2016). Upon ISRIB administration to wild-type larvae with hematopoietic progenitors marked with *tepIV-GFP*, there was a significant increase in progenitor differentiation into plasmatocytes (Fig. S4B,C), crystal cells (Fig. S4E,F) and about 80% LGs positive for lamellocyte production (Fig. S4H,I) when compared to its vehicle control (Fig. S4A,D,G-G′,C,F,I).

Since ISR is activated upon induction of Toll or Imd pathway in the hematopoietic progenitors we hypothesized that larvae where the progenitors encounter chronic Toll or Imd upregulation when subjected to sustained activation of ISR pathway can potentially counter the stress conditions. Such aged progenitors where cellular homeostasis is disrupted when exposed to sustained ISR pathway activation could possibly counter the stress and maintain hematopoiesis. In order to test this hypothesis, we treated larvae bearing Toll or Imd activation in the hematopoietic progenitors with Histidinol, an ISR pathway activator. Histidinol treatment to larvae with Toll activation in the progenitors leads to a rescue of plasmatocyte differentiation phenotype (Fig. S4K,L) whereas crystal cell numbers show no difference (Fig. S4N,O) as compared to the vehicle treatment to larvae with progenitor- specific Toll activation (Fig. S4J,M,L,O). There was also a moderate decrease in the percentage of LGs positive for lamellocytes upon Histidinol treatment (Fig. S4Q-Q′,R) as compared to the vehicle control (Fig. S4P-P′,R). Similarly, we treated the larvae with Imd activation in progenitors with Histidinol which resulted in a rescue in both plasmatocyte (Fig. S4T,U) as well as crystal cell differentiation (Fig. S8W,X). In this case, the percentage of LGs positive for lamellocytes also decreased (Fig. S4Z-Z″) upon Histidinol treatment as compared to the vehicle control (Fig. S4Y-Y′,Z″) where larvae with Imd activation in progenitors are treated with the vehicle alone.

## Ectopic hyperactivation of the ISR pathway in a disrupted cellular homeostasis scenario restores LG hematopoiesis

Our pharmacological intervention data suggests that sustained ISR activation in Toll or Imd activated background can rescue blood cell homeostasis. To validate if this holds true upon genetic activation of ISR pathway in Imd activated background, we over-expressed the activated form of Gcn2 or CrcRA (a splice variant and isoform of *crc* gene that encodes *Drosophila* Atf4) (Vasudevan et al., 2022) in the progenitors where Imd is activated to check if the hematopoiesis parameters are altered. Over-expression of *gcn2^{act}* or *crcRA* in progenitors where Imd is over-activated leads to a rescue of plasmatocyte (Fig. 7B,C,D) and crystal cell differentiation (Fig. 7F,G,H) as compared to the control where Imd is over-activated in progenitors (Fig. 7A,D,E,H). We do not find any significant rescue of the lamellocyte phenotype upon over-expression of *gcn2^{act}* or *crcRA* upon Imd over-activation (Fig. 7J-K′,L) as

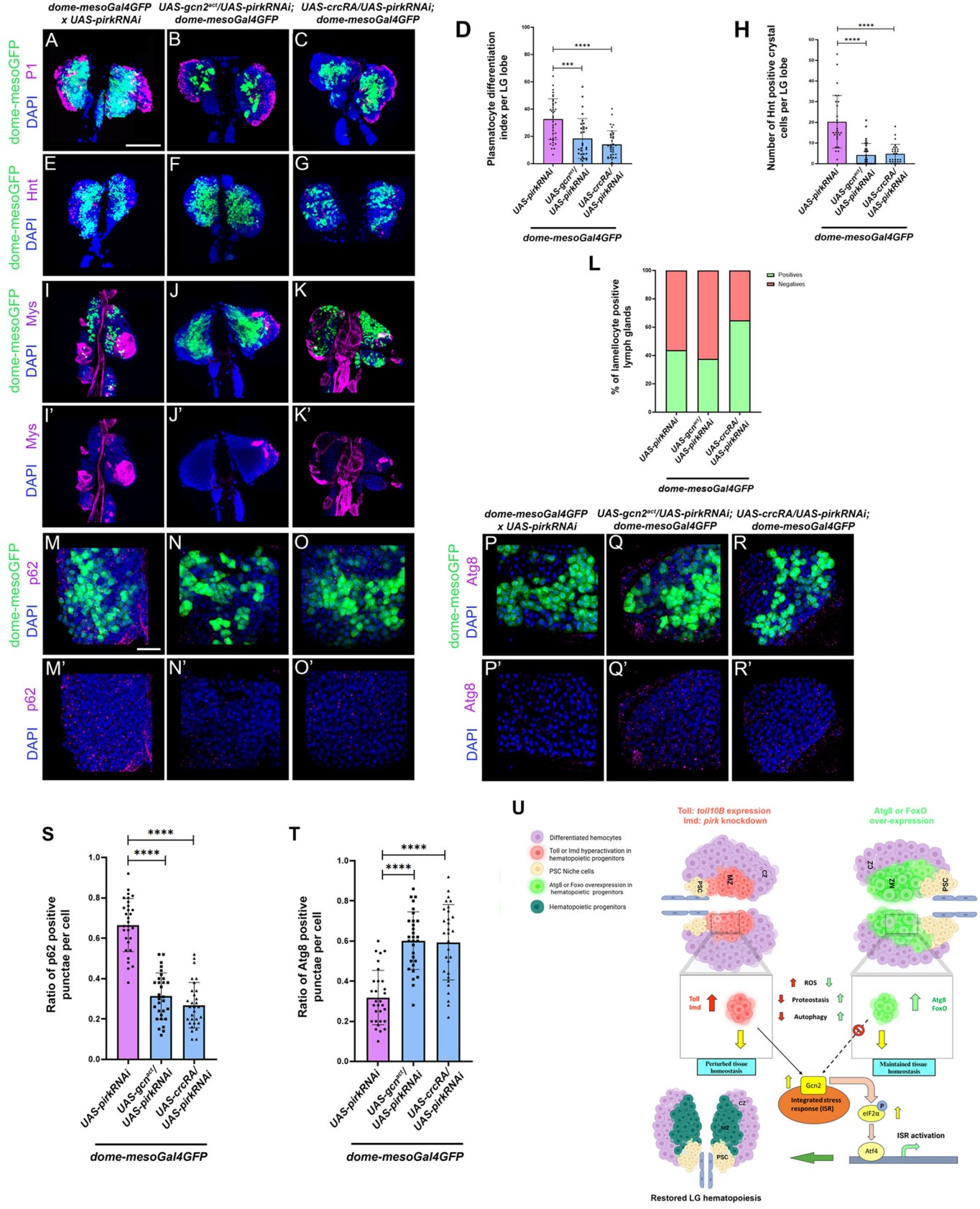

**Fig. 7.** See next page for legend.

**Fig. 7. Sustained upregulation of ISR pathway in Imd over-activation scenario in hematopoietic progenitors is capable of restoring hematopoiesis.** Plasmatocyte differentiation (P1, magenta) or crystal cell differentiation (Hnt, magenta) or lamellocyte differentiation (Mys, magenta) or evaluation of p62 or Atg8 positive punctae per cell marked by p62 or Atg8 antibody (magenta) upon distal progenitor-specific (using *dome-mesoGal4*) hyperactivation of components of ISR pathway viz. *gcn2act* and *crcRA* in an IMD upregulated background (via *pirkRNAi*) as compared to wild-type control (A-K′,M-O′,P-R′). Graphical representation of plasmatocyte differentiation index or number of crystal cells or percentage of lamellocyte positive LGs or ratio of p62 or Atg8 positive punctae per cell upon *dome-mesoGal4* mediated over-expression of *gcn2act* and *crcRA* in a *pirk* knockdown background as compared to wild-type control (D,H,L,S,T). *UAS-gcn2act/UAS- pirkRNAi; dome-mesoGal4* and *UAS-crcRA/UAS-pirkRNAi; dome-mesoGal4* were analyzed for plasmatocyte differentiation (*gcn2act*: N=15, n=30; *crcRA*: N=19, n=37) or crystal cell numbers (*gcn2act*: N=17, n=33; *crcRA*: N=14, n=28) or lamellocyte differentiation (*gcn2act*: N=16, n=32; *crcRA*: N=15, n=30) or p62 levels (*gcn2act*: N=11; *crcRA*: N=10) or Atg8 levels per cell (*gcn2act*: N=10; *crcRA*: N=12) as compared to wild-type control (plasmatocytes: N=19, n=38; crystal cells: N=14, n=28; lamellocytes: N=15, n=30; p62: N=12; Atg8: N=10). N, number of larvae; n, number of individual primary LG lobes. Individual data points in the graphs represent individual primary lobes of the LG. GFP (green) is driven by *dome-mesoGal4* (A-K′,M-O′,P-R′). Nuclei, DAPI (blue). Values are mean±s.d. and asterisks denote statistically significant differences (ns denotes not significant, **$P<0.01$, ***$P<0.001$, ****$P<0.0001$). Student's *t*-test with Welch's correction was performed for the statistical analysis. Our working model summarizing how restoration of LG hematopoiesis occurs upon ectopic ISR upregulation in the hematopoietic progenitors in an Imd pathway over-activation genetic background (U). Created in BioRender. Khadilkar, R. J., (2025). https://BioRender.com/w52d960. This figure was sublicensed under CC-BY 4.0 terms. Scale bar: 50 µm (A-K′), 30 µm (M-R′).

compared to the respective control (Fig. 7I-I′,L). These results indicate that both chemical and genetic ISR activation are capable of restoring blood cell homeostasis in LGs where Toll or Imd is activated in hematopoietic progenitors. In addition to the hematopoietic parameters, we also assessed if quality control mechanisms like autophagy that are perturbed upon disruption of cellular homeostasis are restored upon ISR activation. As per our previous results we had shown that autophagy is affected in Toll or Imd activated scenario (Fig. 1). Here, we over-expressed *gcn2act* or *crcRA* in the progenitors where Imd is over-activated and scored for p62. Our analysis indicates that p62 positive puncta per cell are lower upon *gcn2act* or *crcRA* over-expression in the Imd over-activated background in the progenitors (Fig. 7N-O′,S) as compared to the control (Fig. 7M-M′,S) whereas Atg8 shows an opposite trend as compared to the respective control which shows that autophagy levels are restored upon ISR activation in the Toll or Imd activated scenario (Fig. 7P-R′,T). Our findings demonstrate that genetic perturbation of cellular homeostasis has distinct effects on LG hematopoiesis and ISR pathway is crucial for responding to stress inflicted by chronic over-activation of Toll or Imd signaling. Sustained ISR activation is capable of countering the stress and restoring blood cell homeostasis in the LG (Fig. 7U).

## DISCUSSION

One of the key hallmarks of aging is stem cell exhaustion where the ability of stem cell self-renewal is compromised leading to aberrant differentiation (López-Otín et al., 2023; Yi et al., 2020). Here, using *Drosophila* hematopoisis as our model we carried out a systematic characterization of how genetic modulation of the cellular homeostasis can impact overall organ homeostasis. We not only modulate cellular homeostasis locally but also investigate how systemic modulation can regulate hematopoiesis via inter-organ communication. We use two genetic approaches – Toll or Imd over-activation for disrupting cellular homeostasis and Foxo or Atg8

over-expression for maintaining or reinstating cellular homeostasis. We supplement these data by chemical-based intervention approach using Bortezomib that induces loss of proteostasis by inhibiting the proteasomal system or by Rapamycin that inhibits mTORC1 thereby activating anti-aging mechanisms. Our results demonstrate that LG homeostasis in terms of niche cell numbers and the extent of blood cell differentiation is abrogated when Toll or Imd pathways are over-activated. Our observations indicate that in Toll or Imd over-activation background, the blood progenitors display the typical features like loss of protein turnover, deregulation of autophagy and increase in ROS levels. We find that the cells undergoing abrogated cellular homeostasis are capable of sensing cellular stress and respond to it by switching on the ISR pathway. Genetic perturbation of ISR components in the LG results in defective hematopoiesis. We hypothesized if sustained activation of the ISR pathway over and above existing levels of activation can restore hematopoiesis since mere switching on of the ISR pathway is not sufficient to restore hematopoiesis to a normal state. Our results indicate that genetic and chemical-based approaches of activation of the ISR pathway in Toll or Imd over-activated scenario are capable of restoring hematopoiesis. Taken together, our results provide novel mechanistic insights into the signaling biology underlying aging at the cellular level during hematopoiesis.

We first investigated if the niche size is affected upon localized and systemic modulation of cellular homeostasis. Niche size reduces upon Toll pathway activation in the niche, MZ or in the fat body. It was earlier reported that Toll pathway hyperactivation in the niche leads to permeability barrier breakdown triggering hemocyte differentiation (Khadilkar et al., 2017). Atg8 over-expression has no effect on the niche size whereas Foxo over-expression cell autonomously in the PSC, whole LG or systemically in the fat body reduces niche size possibly because Foxo is a downstream effector and feedback regulator of insulin signaling (Puig and Tjian, 2005) that attenuates insulin signaling (Ni et al., 2007). Insulin signaling plays an important role in regulating PSC size in the LG (Benmimoun et al., 2012) and Foxo over-expression dysregulates insulin pathway (Ni et al., 2007). There is increased plasmatocyte differentiation in both Toll and Imd activated background whereas crystal cell differentiation decreases upon Toll activation in the niche. Previous reports suggest that crystal cell lineage differentiation needs to be inhibited in order to promote lamellocyte differentiation (Small et al., 2013; Deichsel et al., 2023). Here, Toll activation inhibits crystal cells promoting lamellocyte differentiation. However, it needs to be now tested if Toll pathway activation in the niche inhibits Notch activity promoting lamellocyte differentiation. Atg8 over-expression has no effect on blood cell differentiation whereas Foxo inhibits crystal cell lineage, promotes plasmatocyte differentiation and shows lamellocyte differentiation in some LGs likely due to inhibition of Insulin signaling as is the case for example, for Insulin receptor (InR) knockdown in the niche that promotes plasmatocyte differentiation (Benmimoun et al., 2012). Domeless progenitor specific Toll or Imd over-activation triggers hemocyte differentiation whereas there is suppression of crystal cells in case of Foxo over-expression and promotes production of lamellocytes in few LGs whereas no effect on differentiation at all upon Atg8 over-expression. *chizGal4* mediated modulation of cellular homeostasis triggers plasmatocyte, crystal cells and lamellocyte differentiation in the Imd activated background and a decrease in crystal cell differentiation in the Toll activated background possibly due to inhibition of Notch to promote lamellocyte production as has been discussed earlier. Upon Foxo or Atg8 over-expression using *chizGal4*, there is no effect on hemocyte differentiation of all three lineages.

Systemic modulation of cellular homeostasis impacts LG hematopoiesis which is in line with previous literature demonstrating systemic regulation of hematopoiesis (Benmimoun et al., 2012; Shim et al., 2012; Yang et al., 2015; Cho et al., 2018; Koranteng et al., 2022). Now, loss of Atg8a in the muscles affects autophagy, reduces lifespan and muscle integrity (Xu et al., 2024). Similarly, Foxo over-expression in muscles (Demontis and Perrimon, 2010) and fat body (Hwangbo et al., 2004) was shown to extend lifespan. Additionally, chronic Imd activation in the fat body resulted in shorter lifespan in the absence of infection (Sciambra and Chtarbanova, 2021). Based on these reports, we selected muscle as a distant organ to investigate the inter-organ communication with the LG. Our results with muscle-specific *mhcGal4* show that Toll or Imd over-activation triggers differentiation whereas Foxo or Atg8 over-expression suppresses differentiation or has no significant effect indicating that both localized and systemic modulation of cellular homeostasis is capable of altering hematopoiesis.

Alternative chemical intervention approaches validate our findings. Bortezomib is an inhibitor of the ubiquitin- mediated proteasomal pathway (Chen et al., 2011) thereby resulting in increased protein instability, redox imbalance and accelerated aging (Manola et al., 2019). Also, we used Rapamycin as previous literature has shown that Rapamycin inhibits mTORC1 and this inhibition of mTORC1 kinase by Rapamycin leads to upregulation of autophagy, increased resistance to starvation and lifespan extension in *Drosophila* (Bjedov et al., 2010). Bortezomib induces whereas Rapamycin suppresses hemocyte differentiation. We find that Rapamycin is capable of rescuing the Imd activation phenotypes of increased plasmatocyte and crystal cell differentiation but not the lamellocyte differentiation phenotype. Rapamycin was unable to rescue the Toll over-activation phenotypes of increased differentiation which needs further mechanistic investigation to tease out why Rapamycin is not capable of suppressing Toll over-activation mediated hemocyte differentiation.

Since cellular stressors like impaired autophagy, elevated ROS levels, loss of protein turnover are sensed by signaling pathways like the ISR pathway (Kroemer et al., 2010; Costa-Mattioli and Walter, 2020; Giardin et al., 2020; Ulfig and Jakob, 2024) we were curious to check if ISR is activated upon Toll or Imd over-activation. We investigated *gcn2* transcript levels in blood cells upon modulation of cellular homeostasis and also looked at peIF2α levels in the LG as an ISR activation readout. Our analysis indicates that *gcn2* mRNA levels are higher upon Toll or Imd over-activation consequently resulting in peIF2α higher levels in the LG. We debated why hematopoiesis is not restored even upon ISR pathway activation in Toll or Imd over-activation scenario. To understand the role of ISR pathway components in the LG, we perturbed ISR components in the hematopoietic progenitors and find that depletion of ISR components results in increased blood cell differentiation whereas over-expression of *gcn2* or *atf4* has either no effect or suppresses hemocyte differentiation. Also, *gcn2* null or *crc* heterozygous mutants display increased hemocyte differentiation aligning with data obtained upon depletion of ISR components in progenitors. We chemically inhibited the ISR pathway using ISRIB, a known inhibitor of the ISR pathway (Zyryanova et al., 2020) and activated it using Histidinol (Taniuchi et al., 2016). ISRIB induces hemocyte differentiation indicating that the ISR pathway is important for homeostasis. We then tested if phenotypes observed upon Toll or Imd over-activation in progenitors are rescued by Histidinol treatment. Histidinol can more potently rescue Imd over-activation phenotypes whereas only plasmatocyte differentiation phenotype upon Toll over-activation can be rescued by Histidinol but not others. The mechanistic link between Histidinol mediated ISR

activation and Toll or Imd pathway warrant further investigation. Lastly, since we debated that existing activation levels of ISR cannot restore physiological hematopoiesis upon Toll or Imd over-activation, we hypothesized if sustained activation of ISR over and above existing levels in this background can restore hematopoiesis. *gcn2* or *atf4* over-expression in *pirk* depletion background can restore the increased plasmatocyte or crystal cell differentiation whereas it cannot rescue the increased lamellocyte production. Furthermore, we found that autophagy levels are restored upon ectopic over-activation of ISR in Imd activated background.

ISR plays a critical role in combating stress especially in Toll or Imd over-activation background and is crucial for ameliorating the detrimental effects of disrupted cellular homeostasis. Our work paves way to investigate if aged HSCs and their niche microenvironment responds to age associated stressors similarly and whether the mechanisms bear similarity or not in the mammalian system. Taken together, aging associated changes impact at the cellular level that have a spiraling effect on the overall health and lifespan. Understanding the mechanisms of how cellular homeostasis is affected as age progresses will help in devising novel strategies to tackle cellular aging.

## MATERIALS AND METHODS
### *Drosophila* genetics
All the *Drosophila* stocks and crosses were maintained at 25°C, in a standard cornmeal diet containing corn starch and sugar as carbon source, malt extract containing trace number of vitamins and minerals, yeast extract as protein source and agar as solidifying agent. *Canton-S* was used as wild-type control. Tissue specific *Gal4* promoter line was used to drive the expression of *UAS* responder genes. Respective *UAS* or *Gal4* parent stocks or *Canton-S* were used as controls wherever appropriate. *Gal4* driver lines used were *collierGal4* driving *UAS-mCD8GFP* (gift from Dr. Michele Crozatier and Dr. Lucas Waltzer, Toulouse, France*), tepIVGal4GFP* on Chr. II*, domeGal4GFP* on X*, hmlΔGal4GFP* on Chr. II (gift from Dr. Lucas Waltzer, Toulouse, France), *chizGal4GFP* on Chr. II (gifted by Dr. Bama Charan Mondal, BHU, India), *dome-mesoGal4GFP* on Chr. III, *hmlGal4GFP* on Chr. II, *e33cGal4* on Chr. III (gifted by Dr. Maneesha Inamdar, JNCASR & DBT-InStem) and muscle specific *mhcGal4* on Chr. III, (RRID:BDSC_55133) driving *UAS-mCD8GFP* (Chr. II*,* RRID: BL_5137). The *UAS*-transgene lines used were *UAS-pirkRNAi* (Chr. II, RRID: BL_67011), *UAS-toll10B* (Chr. X, RRID: BDSC_58987), *UAS-atg8GFP* (Chr. III, RRID:BDSC_51656), *UAS-foxoGFP* (Chr. III, RRID: BDSC_43633), *UAS-gcn2^{act}, UAS-crcRA/cyO, crc^{1/+}* on II, *gcn2^{FRT\ 12kb\ -/-}* on Chr III, *gcn2^{wt}* rescue; *gcn2^{FRT\ 12kb\ -/-}* (gifted by Dr. Hyung Don Ryoo, New York University Grossman School of Medicine, Kang et al., 2016), *UAS-gcn2RNAi* (Chr. II, RRID: VDRC_103976), *UAS-eif2αRNAi* (Chr. II, RRID: VDRC_V104562), *UAS-atf4RNAi* (Chr. II, RRID: VDRC_109014).

### Antibodies
Antibodies used were mouse-raised anti-P1 (1:100, kind gift from Dr. Istvan Ando), mouse-raised anti-Hindsight (1:25, 1G9 – DSHB; RRID: Ab_528278), mouse-raised anti-Antp (1:25, 8C11- DSHB; RRID: Ab_528083), mouse-raised anti-Myospheroid (1:25, 6G11 - DSHB; RRID: Ab_528310), rabbit-raised anti-p-eIF2α (1:100, Cell Signaling Technology, RRID:Ab_119A11), CellROX Deep Red Reagent (2.5 mM in DMSO, Invitrogen, C10422), PROTEOSTAT Protein detection assay (1:100, Enzo Life Sciences, cat. no.: ENZ-51023-KP050), rabbit-raised anti-P62/SQSTM1 (1:250, Proteintech, RRID:AB_10694431) and rabbit-raised anti-ATG8 (1:200, Sigma-Aldrich, RRID:Ab_2939040) for immunofluorescence based experiments. Normal goat serum (HIMEDIA, RM10701) was used as the blocking agent. Alexa-Fluor 568 conjugated secondary antibodies – goat-raised anti-mouse 568 (1:400, Invitrogen, RRID: Ab_144696), goat-raised anti-mouse 633 (1:400, Invitrogen, RRID: AB_2535719) and goat-raised anti-rabbit 568 (1:400, Invitrogen, RRID: AB_10563566) were used for immunofluorescence-based experiments.

## LG dissection, immunohistochemistry and mounting

Wandering late third instar larvae were used for LG dissections. The dissections were performed in phosphate buffer saline (PBS), fixed in 4% paraformaldehyde for 20 min, followed by three washes with PBS containing 0.3% Triton-X (PBST) for 5 min each. The samples were then blocked in 20% normal goat serum for 20 min at room temperature followed by overnight primary antibody incubation at 4°C. This was followed by PBST washes, blocking, and treatment with appropriate Alexa-Fluor conjugated secondary antibody incubation for 2 h at room temperature. This was again followed by three PBST washes for 5 min each. The LGs were then mounted in Vectashield mounting medium containing DAPI (Vector Laboratories, RRID: AB_2336790).

## Chemical treatment

For drug treatments, early- to mid-third instar larvae were collected and transferred to vials without food containing few drops of distilled water (to prevent cuticle desiccation) and starved for 2 h. The larvae were then transferred to food containing corresponding drugs to be used for treatment. Chemicals used include Rapamycin (40 µg/ml, R0395, Sigma-Aldrich) dissolved in absolute ethanol, Bortezomib (10 µM, 5.04314.2201, EMD Millipore) dissolved in DMSO (50 ml, D8418, Sigma-Aldrich), ISRIB (5 nm, SML0843, Sigma-Aldrich) dissolved in DMSO and Histidinol (4 mM, H6647, Sigma-Aldrich) dissolved in Nuclease free water (500 ml, 1097715, Invitrogen). For control, larvae were fed on food containing the respective solvent (vehicle) alone, post starvation. For each of the drug treatment experiments, at least 15 larvae were used for analysis. LGs from the treated larvae for both the chemical and vehicle treatment were dissected 14 to 16 h post treatment.

## Image acquisition and analysis of various LG parameters

Images were acquired on Zeiss LSM 780 or Leica SP8 confocal microscope. LG images were acquired using the 40x objective on both LSM 780 (40X objective, 1.4 NA, oil immersion) and Leica SP8 (40X objective, 1.3 NA, oil immersion) confocal microscopes using either 512×512 or 1024×1024 frame size with no averaging and a z-step size of 1.5 or 2.5 microns. Images were analyzed for various hematopoietic parameters or for analysis of ROS levels, levels of autophagy, protein turn-over and p-eIF2α levels. The raw images were processed into .tiff file format (in RGB mode) into single or merge channels as per requirement using ImageJ. These .tiff files were then opened using Adobe Photoshop (Creative Cloud version) and were assembled and stitched as per requirement of the figure panel using a black background on a Photoshop canvas (RGB mode). Final assembled figure panels were saved in TIFF format using LZW compression with a resolution of 600 dpi or more.

## Image analysis

### Hematopoietic parameters

Confocal images were captured using either Zeiss LSM 780 or Leica SP8 confocal microscope. Z projection of the confocal images was used for estimating various LG parameters using ImageJ/Fiji software. Plasmatocyte differentiation Index was estimated by measuring the percentage of P1 positive area divided by the total area of the LG primary lobe. The prohemocyte Index was estimated by measuring the percentage of *tepIV-GFP* or *dome-GFP* positive area divided by the total area of LG primary lobe. Freehand selection tool was used for measuring the area of the plasmatocytes or the prohemocytes. For the quantitation of Antp and Hnt, the positive signals for respective markers were manually counted using the multipoint tool. For quantitation of Mys, the LGs were categorized as positive or negative based on presence or absence of lamellocytes and percentage of lamellocyte positive LGs was estimated. The LG quantitations were done for individual primary LG lobes.

### Analysis of expression

Confocal images were captured using either Zeiss LSM 780 or Leica SP8 confocal microscope. For estimation of ROS levels using CellROX Deep Red Reagent in *tepIVGal4*-specific core-progenitor population in the LG using ImageJ/Fiji software (RRID:SCR_003070), the images for *tepIVGal4GFP×wt, tepIVGal4GFP×UAS-toll10B, tepIVGal4GFP×UAS-*

*atg8GFP & tepIVGal4GFP×UAS-foxoGFP* were acquired at the same intensity settings/parameters and mean fluorescent intensity (represented as arbitrary units) in the progenitors was calculated keeping the threshold value of fluorescent intensity uniform as that of *tepIVGal4GFP×wt* for all the above crosses. Similar protocol was followed for estimating the mean fluorescence intensity of p-eIF2α in *tepIVGal4GFP×wt* and *tepIVGal4GFP×UAS-pirkRNAi, tepIVGal4GFP×UAS-toll10B, tepIVGal4GFP×UAS-atg8GFP and tepIVGal4GFP×UAS-foxoGFP*, which was calculated keeping the threshold value of fluorescent intensity uniform as that of *tepIVGal4GFP×wt* for all the above crosses.

For the quantitation of proteotoxic stress and autophagic flux, we unbiasedly selected several fields from each image and within each field we counted the total number of punctae ($P$) for Proteostat, p62 and Atg8 respectively and also counted the total number of nuclei ($n$). In our quantitation, we took the ratio of p/n, which is representative of puncta per cell considering all LG cells are uninucleate. Thus, we compared the Proteostat or p62 or Atg8 puncta per cell in the LG of different experimental sets.

## Quantitative real-time PCR

Hemolymph was extracted from 300 larvae mounted in cold PBS by puncturing the cuticle using minutien insect pins. The hemolymph was pelleted by centrifuging at 2000 $g$ for 7 min at 4°C. The supernatant was removed and the hemolymph pellet was lysed in TRIzol (Ambion, Life Technologies, cat. no.: 11596018). The lysates were stored at −80°C. Hemolymph collection were done in batches of 50 to 100 larvae and once the hemolymph from all 300 larvae were done, RNA was isolated by pooling all the aqueous layers post-chloroform treatment, followed by RNA isolation according to the manufacturer's protocol. RNA yield was quantified using Nanodrop. 1 µg of mRNA was reverse transcribed using oligo-dT primers (Promega, C110A) and ImProm-II (Promega, A3800). Quantitation of the mRNA transcripts was done using SYBR green chemistry (Thermo Fisher Scientific, cat. no.: 4367659) in the Quantstudio 5 RT PCR system (Thermo Fisher Scientific) in quadruplets of 10 µl reaction. The data were analyzed using the ΔΔCt method and relative mRNA expression was normalized to *rp49*. Fold change calculations were done in comparison to wild-type control. The experiment was done in biological triplicates and statistical analysis was performed using one-way ANOVA (Dunnett) for comparison of all test genotypes with wild-type control genotype.

## List of primers

qRT primers in 5′ to 3′ direction
*gcn2* Forward CCAACGGACATACGGATACAAC
*gcn2* Reverse CGTAGCTCTTGGGATTGAGCC
*rp49* Forward GCTAAGCTGTCGCACAAATG
*rp49* Reverse GTTCGATCCGTAACCGATGT

## Genotypes and genetic crosses

In Fig. 1, to evaluate the conventional hallmarks of cellular aging, cellular homeostasis aspects were observed including estimation of autophagy levels, ROS levels and proteostasis regulation via protein turnover assessment in the *tepIV*-positive core progenitors, the following crosses were set up. For autophagy levels assessment, *tepIVGal4GFP×UAS-pirkRNAi* and *tepIVGal4GFP×UAS-atg8GFP* were compared with *tepIVGal4GFP×wt* as wild-type control. For ROS levels estimation, *tepIVGal4GFP×UAS-toll10B, tepIVGal4GFP×UAS-atg8GFP* and *tepIVGal4GFP×UAS-foxoGFP* were compared with *tepIVGal4GFP x wt* as wild-type control. For assessment of proteostasis via protein turnover estimation, *tepIVGal4GFP×UAS-pirkRNAi* and *tepIVGal4GFP×UAS-toll10B, tepIVGal4GFP×UAS-atg8GFP* and *tepIVGal4GFP×UAS-foxoGFP* were compared to *tepIVGal4GFP×wt* as wild-type control.

In Fig. 2, the following crosses were set up to study the effect of modulation of cellular homeostasis on PSC niche size by localised or systemic cellular subsets of LG:

a.  PSC niche-specific: collierGal4GFP × UAS-pirkRNAi and collierGal4GFP × UAS-toll10B (disruption of cellular homeostasis),

collierGal4GFP × UAS-atg8GFP and collierGal4GFP × UAS-foxoGFP (maintenance of cellular homeostasis) and collierGal4GFP×wt (control).

b. Distal progenitor-specific: domeGal4GFP × UAS-pirkRNAi and domeGal4GFP × UAS-toll10B (disruption of cellular homeostasis), domeGal4GFP × UAS-atg8GFP and domeGal4GFP × UAS-foxoGFP (maintenance of cellular homeostasis) and domeGal4GFP×wt (control).

c. CZ differentiated hemocyte-specific: hmlΔGal4GFP × UAS-pirkRNAi and hmlΔGal4GFP × UAS-toll10B (disruption of cellular homeostasis), hmlΔGal4GFP × UAS-atg8GFP and hmlΔGal4GFP × UAS-foxoGFP (maintenance of cellular homeostasis) and hmlΔGal4GFP×wt (control).

d. Whole LG-specific: e33cGal4 × UAS-pirkRNAi and e33cGal4 × UAS-toll10B (disruption of cellular homeostasis), e33cGal4 × UAS-atg8GFP and e33cGal4 × UAS-foxoGFP (maintenance of cellular homeostasis) and e33cGal4 × wt (control).

e. Systemic fat body-specific: fbGal4 × UAS-pirkRNAi and fbGal4 × UAS-toll10B (disruption of cellular homeostasis), fbGal4 × UAS-atg8GFP and fbGal4 × UAS-foxoGFP (maintenance of cellular homeostasis) and fbGal4 × wt as wild-type control.

In Fig. 3, the following crosses were set up to study the effect of genetic modulation of cellular homeostasis in distal-progenitor population using domeGal4 on LG hematopoiesis. Disruption of cellular homeostasis (domeGal4GFP × UAS-pirkRNAi and domeGal4GFP × UAS-toll10B) and maintenance of cellular homeostasis (domeGal4GFP × UAS-atg8GFP and domeGal4GFP × UAS-foxoGFP) as compared to domeGal4GFP×wt as wild-type control.

In Fig. 4, the following crosses were set up to study the effect of genetic modulation of cellular homeostasis systemically in muscles using mhc-Gal4 on LG hematopoiesis by disruption of cellular homeostasis (mhcGal4GFP × UAS-pirkRNAi and mhcGal4GFP × UAS-toll10B) and maintenance of cellular homeostasis (mhcGal4GFP × UAS-atg8GFP and mhcGal4GFP × UAS-foxoGFP) as compared to mhcGal4GFP×wt as wild-type control.

In Fig. 5, the following crosses were set up to study the systemic effect of chemical modulation of ageing on LG hematopoiesis upon treatment with Bortezomib on tepIVGal4GFP x wt larvae and compared with its solvent (vehicle) control. Similarly, Rapamycin treatment on wild-type larvae (tepIVGal4GFP×wt) and in a scenario where Toll or Imd is activated in core progenitors to investigate its effect on rescue of differentiation (tepIVGal4GFP × UAS-pirkRNAi and tepIVGal4GFP × UAS-toll10B) when compared with its solvent (vehicle) control.

In Fig. 6, the following crosses were set up to check for the expression levels of gcn2 in the hemocyte population (using hmlGal4) and p-eIF2α in core-progenitors (using tepIVGal4). For estimation of gcn2 mRNA transcript levels in all hemocytes, following crosses were set up: hmlGal4GFP × UAS-pirkRNAi and hmlGal4GFP × UAS-toll10B, hmlGal4GFP × UAS-atg8GFP and hmlGal4GFP × UAS-foxoGFP were compared to hmlGal4GFP×wt as wild-type control and for estimation of p-eIF2α in core-progenitors, tepIVGal4GFP × UAS-pirkRNAi and tepIVGal4GFP × UAS-toll10B, tepIVGal4GFP × UAS-atg8GFP and tepIVGal4GFP × UAS-foxoGFP were compared to tepIVGal4GFP×wt as wild-type control. Genetic modulation of ISR pathway via knockdown (UAS-gcn2RNAi, UAS-eIF2αRNAi, UAS-atf4RNAi) or overexpression (UAS-gcn2act and UAS-crcRA) of ISR components in tepIVGal4-positive core progenitors and its effect on blood cell differentiation in LG was studied by setting up the following crosses: tepIVGal4GFP × UAS-gcn2RNAi, tepIVGal4GFP × UAS-eIF2αRNAi, tepIVGal4GFP × UAS-

atf4RNAi, tepIVGal4GFP × UAS-gcn2act, tepIVGal4GFP × UAS-crcRA as compared to tepIVGal4GFP×wt as wild-type control.

In Fig. 7, ISR components including gcn2act and crcRA were ectopically over-expressed in distal-progenitor population using dome-mesoGal4GFP in IMD pathway upregulated (pirkRNAi) background (resultant genotype: UAS-gcn2act or UAS-crcRA/UAS-pirkRNAi; dome-mesoGal4GFP/dome-mesoGal4GFP or dome-mesoGal4GFP/TM6B) and compared with dome-mesoGal4GFP × UAS-pirkRNAi as control to investigate blood cell homeostasis and status of autophagy upon ISR activation in Imd activated genetic background.

In Fig. S1, the following crosses were set up to study the effect of modulation of cellular homeostasis on blood cell differentiation upon induction of Toll or Imd pathway (UAS-pirkRNAi and UAS-toll10B) and maintenance of cellular homeostasis (UAS-atg8GFP and UAS-foxoGFP) in PSC niche using collierGal4 - collierGal4GFP × UAS-pirkRNAi, collierGal4GFP × UAS-toll10B, collierGal4GFP × UAS-atg8GFP, collierGal4GFP × UAS-foxoGFP and collierGal4GFP×wt as wild-type control.

In Fig. S2, the following crosses were set up to study the effect of modulation of cellular homeostasis on hemocyte differentiation in LG upon induction of Toll or Imd pathway (UAS-pirkRNAi and UAS-toll10B) and Foxo or Atg8 over-expression (UAS-atg8GFP and UAS-foxoGFP) in intermediate-progenitor population using chizGal4: chizGal4GFP × UAS-pirkRNAi, chizGal4GFP × UAS-toll10B, chizGal4GFP × UAS-atg8GFP, chizGal4GFP × UAS-foxoGFP and chizGal4GFP×wt as wild-type control.

In Fig. S3, to study the effect of ISR mutants on hemocyte differentiation in LG, the following lines were used: gcn2^{FRT 12kb −/−} null mutant and gcn2^{wt} rescue; gcn2^{FRT 12kb −/−} as its control and crc^{1/+} hypomorphic allele and a wild-type control.

In Fig. S4, the following crosses were set up to study the effect of chemical modulation of ISR pathway on LG homeostasis upon treatment with ISRIB on tepIVGal4GFP x wt larvae compared with its solvent (vehicle) control. Similarly, Histidinol treatment under accelerated ageing background (tepIVGal4GFP × UAS-pirkRNAi & tepIVGal4GFP × UAS-toll10B) was performed to investigate its effect on rescue of differentiation and compared with its solvent (vehicle) control.

In Fig. S5 genotypes are same as Fig. 1 and in Fig. S6 genotypes are same as Fig. 2.

## Statistical analysis

Immunofluorescence based experiments and their analysis was performed on at least ten LGs dissected from wandering late third-instar larvae. The sample size for each of the experiments and the 'N' and 'n' values for all genotypes that denote the number of larvae and number of individual primary LG lobes analyzed per genotype, respectively, have been indicated in respective figure legends. Statistical analysis was performed using the GraphPad Prism Version 10 software (RRID:SCR_002798). For analysis of statistical significance, each experimental sample was tested with its respective control in a given experimental setup for all the data in each of the figures in order to estimate the $P$-value. $P$-values were determined by using a two-tailed unpaired Student's $t$-test with Welch's correction. Values are mean±s.d., and asterisks denote statistically significant differences (ns denotes not significant, $*P<0.05$, $**P<0.01$, $***P<0.001$, $****P<0.0001$). Mutant genotypes were compared to the wild-type controls and the knockdown or over-expression genotypes were compared to their respective parental Gal4 controls that were crossed to wild type for all the statistical analysis performed. The experiments where chemical treatment was given were compared to the respective vehicle controls. No statistical method was used to pre-determine the sample size and the experiments were not randomized.

List of genetic models/mutants used in the study and their purpose:

| *Gal4s* or mutants used | Tissue/organ targets | Purpose used for |
|---|---|---|
| *collierGal4* | PSC niche-specific | 1. **Disruption of homeostasis** <br> a. *collierGal4GFP×UAS-pirkRNAi*(Imd pathway hyperactivation) <br> b. *collierGal4GFP×UAS-toll10B*(Toll pathway constitutive activation) <br> 2. **Maintenance of homeostasis** <br> a. collierGal4GFP×UAS-atg8GFP(Atg8 overexpression and autophagy upregulation) <br> b. *collierGal4GFP×UAS-foxoGFP*(Foxo overexpression) |
| *tepIVGal4GFP* | Core progenitor-specific | I. Systemic modulation of cellular homeostasis upon chemical treatment with Rapamycin and Bortezomib <br> II. Investigating rescue of differentiation upon Rapamycin treatment in larvae with following genotypes: <br> a. *tepIVGal4GFP×UAS-pirkRNAi* <br> b. *tepIVGal4GFP×UAS-toll10B* <br> III. Estimation of p-eIF2$\alpha$ levels in core-progenitors upon *tepIV-Gal4* mediated expression of transgenes for disruption or maintenance of homeostasis <br> IV. Genetic modulation of ISR pathway components (knockdown or over-expression) in core progenitors and its effect on hemocyte differentiation <br> V. Systemic modulation of ISR pathway upon chemical treatment with ISRIB <br> VI. Histidinol treatment on larvae with tepIVGal4 mediated activation of Toll or Imd pathway |
| *domeGal4* | Distal progenitor-specific | 1. **Disruption of homeostasis** <br> a. *domeGal4GFP×UAS-pirkRNAi* <br> b. *domeGal4GFP×UAS-toll10B* <br> 2. **Maintenance of homeostasis** <br> a. *domeGal4GFP×UAS-atg8GFP* <br> b. *domeGal4GFP×UAS-foxoGFP* |
| *dome-mesoGal4* | Distal progenitor-specific | Overexpression of ISR components like *gcn2^act* and *crcRA* in distal progenitors with a genetic background of hyperactivated IMD pathway to investigate the rescue of differentiation and reinstatement of cellular homeostasis in LG |
| *chizGal4* | Intermediate progenitors-specific | 1. **Disruption of homeostasis** <br> a. *chizGal4GFP×UAS-pirkRNAi* <br> b. *chizGal4GFP×UAS-toll10B* <br> 2. **Maintenance of homeostasis** <br> a. *chizGal4GFP×UAS-atg8GFP* <br> b. *chizGal4GFP×UAS-foxoGFP* |
| *hmlΔGal4* | CZ differentiated hemocytes-specific | 1. **Disruption of homeostasis** <br> a. *hmlΔGal4GFP×UAS-pirkRNAi* <br> b. *hmlΔGal4GFP×UAS-toll10B* <br> 2. **Maintenance of homeostasis** <br> a. *hmlΔGal4GFP×UAS-atg8GFP* <br> b. *hmlΔGal4GFP×UAS-foxoGFP* |
| *hmlGal4* | Pan hemocyte-specific | Estimation of *gcn2* mRNA levels upon genetic modulation of cellular homeostasis |
| *e33cGal4* | Whole LG-specific | 1. **Disruption of homeostasis** <br> a. *e33cGal4×UAS-pirkRNAi* <br> b. *e33cGal4×UAS-toll10B* <br> 2. **Maintenance of homeostasis** <br> a. *e33cGal4×UAS-atg8GFP* <br> b. *e33cGal4×UAS-foxoGFP* |
| *pplGal4* | Fat body-specific | 1. **Disruption of homeostasis** <br> a. *pplGal4×UAS-pirkRNAi* <br> b. *pplGal4×UAS-toll10B* <br> 2. **Maintenance of homeostasis** <br> a. *pplGal4×UAS-atg8GFP* <br> b. *pplGal4×UAS-foxoGFP* |
| *mhcGal4* | Pan muscle-specific | 1. **Disruption of homeostasis** <br> a. *mhcGal4GFP×UAS-pirkRNAi* <br> b. *mhcGal4GFP×UAS-toll10B* <br> 2. **Maintenance of homeostasis** <br> a. *mhcGal4GFP×UAS-atg8GFP* <br> b. *mhcGal4GFP×UAS-foxoGFP* |
| *gcn2^{FRT 12kb−/−} null mutant* and *crc^{1/+} hypomorphic allele* | ISR pathway mutants (whole animal mutants) | Investigating the effect of ISR mutants on hemocyte differentiation in LG |

## Acknowledgements
We would like to thank the Digital Imaging Facility (DIF) and the Common Instrumentation Facilities (CIF) at ACTREC for all the support. We thank the Bloomington *Drosophila* Stock Center, Developmental Studies Hybridoma Bank and the fly community for fly stocks and antibodies. We would like to particularly thank Lucas Waltzer, Hyung Don Ryoo for various fly lines and reagents. We are thankful to the Stem Cell and Tissue Homeostasis lab for useful input and discussions.

## Competing interests
The authors declare no competing or financial interests.

## Author contributions
Conceptualization: R.J.K.; Data curation: R.J.K., K.G., R.K.I., M.S.I.; Formal analysis: K.G., R.K.I., S.S., M.S.I., J.G.L.; Funding acquisition: R.J.K.; Investigation: K.G., R.K.I., S.S., M.S.I., J.G.L.; Methodology: K.G., R.K.I., S.S., M.S.I., J.G.L.; Project administration: R.J.K.; Resources: R.J.K.; Supervision: R.J.K.; Validation: K.G., R.K.I., S.S., M.S.I.; Visualization: R.J.K.; Writing – original draft: R.J.K., K.G., R.K.I.; Writing – review & editing: R.J.K., K.G., R.K.I.

## Funding
We would like to thank the Department of Biotechnology, Ministry of Science and Technology, India for the Har Gobind Khorana – Innovative Young Biotechnologist Award (no. BT/13/IYBA/2020/14) to R.J.K., Ramalingaswami Re-entry Fellowship from the Department of Biotechnology, Ministry of Science and Technology, India (BT/RLF/Re-entry/19/2020) to R.J.K., Council of Scientific and Industrial Research (CSIR)-JRF to K.G. This work was also funded by a Basic and Translational Research in Cancer grant (no.1/3(7)/2020/TMC/R&D-II/8823 Dt.30.07.2021), Capacity Building and Development of Novel and Cutting-edge Research Activities (no.1/3(4)/2021/TMC/R&D-II/15063 Dt.15.12.2021) from the Department of Atomic Energy, Government of India. Open Access funding provided by Advanced Centre for Treatment, Research and Education in Cancer, India. Deposited in PMC for immediate release.

## Data and resource availability
All relevant data can be found within the article and its supplementary information.

## First person
This article has an associated First Person interview with the co-first authors of the paper.

## Peer review history
The peer review history is available online at https://journals.biologists.com/bio/article-lookup/doi/10.1242/bio.062046.reviewer-comments.pdf.

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
