## [Peer Review File · Biology Open]

Genetic perturbation of cellular homeostasis regulates Integrated Stress Response signalling to control *Drosophila* hematopoiesis

Kishalay Ghosh, Rohit Krishnan Iyer, Saloni Sood, Mohamed Sabeelil Islam, Jyotsana Labad and Rohan Khadilkar

DOI: 10.1242/bio.062046

Editor: Alissa Armstrong

Review timeline

Original submission:	10 April 2025
Editorial decision:	19 April 2025
First revision received:	2 May 2025
Editorial decision:	16 May 2025
Second revision received:	2 June 2025
Accepted:	5 June 2025

Original submission

First decision letter

MS ID#: bio.062020

MS Title: Modulation of cellular ageing regulates Integrated Stress Response signalling to control blood cell homeostasis

Authors: Rohan Jayant Khadilkar; Kishalay Ghosh; Rohit Krishnan Iyer; Saloni Sood; Mohammed Sabeelil Islam; Jyotsana G Labad

Dear Dr Khadilkar,

I have now reached a decision on the above manuscript.

The reviewer reports are shown at the bottom of this email or can be accessed, together with a copy of this decision letter, by going to:

Review of your article has raised several important concerns that, together, are significant enough to prevent me from accepting it for publication. I am sorry to write with this disappointing news; however, I am sure that you appreciate that the conclusions of your research must be seen by the wider community to be fully supported by the data acquired using the most appropriate methodology.

Having said that, should you be able to carry out all the work suggested by the referees, then I would be happy to see the paper again, as a new submission. If after considering the feedback, you instead decide to submit elsewhere, please let me know, so that we can close our file.

If you decide to resubmit, please go to:

<https://www.editorialmanager.com/bio/>

and click on the 'Submit a manuscript' link.

Reviewer 1

Comments for the author

In this manuscript, Rohan Jayant Khadilkar and colleagues use *Drosophila* as model system to analyze the potential impact of genetic perturbation of cellular homeostasis on the major hematopoietic organ, the lymph gland (LG). For this purpose, they use a combination of genetic tools (Foxo and Atg8 upregulation to balance cellular homeostasis, and Toll and Imd activation to disrupt cellular homeostasis) and chemicals (Rapamycin to block TOR and activate autophagy and Bortezomib to block the proteasome) to address the impact on the three different cell populations that comprise the LG, namely the niche (PSC), the progenitor cell population and the differentiated hemocytes. They also induce expression of these transgenes in other tissues with well-known systemic effects (eg. fat body or muscle) and see an effect on LG homeostasis. Authors identify a role of the integrated stress response (ISR) in this process, as genetic or chemical manipulation of ISR was able to partially rescue the effects of disrupted cellular homeostasis in LG homeostasis. The paper is well written and present a large collection of data that will be useful to the LG community interested in tissue homeostasis and the role of the ISR in this process.

Minor changes:

(1) Single channels should be shown in Figures 1 and 2

Reviewer 2: Good paper. You switch from ageing to aging. I think it would be uniform to pick one and stick with it throughout the paper.

Reviewer's Responses to Questions

Experimental quality

Does each figure have the proper controls?

If 'No', please indicate reasons in Comments for Author box below.

Reviewer #1:

Yes

Reviewer #2:

Yes

Were the data analyzed using appropriate statistical tests?

If 'No', please indicate reasons in Comments for Author box below.

Reviewer #1:

Yes

Reviewer #2:

Yes

Reproducibility

Were experiments performed using adequate number of biological replicates?

If 'No', please indicate reasons in Comments for Author box below.

Reviewer #1:

Yes

Reviewer #2:

Yes

Does the methods section provide sufficient detail to permit reproducibility?
If 'No', please indicate reasons in Comments for Author box below.

Reviewer #1:

Yes

Reviewer #2:

Yes

Completeness

Are the manuscript's conclusions supported by the data?
If 'No', please indicate reasons in Comments for Author box below.

Reviewer #1:

No

Reviewer #2:

Yes

Scholarship

Do the authors cite and discuss the merits of data that would argue for and against their conclusion?
If 'No', please indicate reasons in Comments for Author box below.

Reviewer #1:

Yes

Reviewer #2:

Yes

Does the manuscript title & abstract accurately reflect the contents of the manuscript, without hyperbole?
If 'No', please indicate reasons in Comments for Author box below.

Reviewer #1:

Yes

Reviewer #2:

Yes

Resubmission
Author response to reviewers' comments

Reviewer 1

In this manuscript, Rohan Jayant Khadilkar and colleagues use *Drosophila* as model system to analyze the potential impact of aging on the homeostasis of the major hematopoietic organ, the lymph gland (LG). For this purpose, they use a combination of genetic tools (Foxo and Atg8 upregulation to induce decelerated aging, and Toll and Imd Response To Reviewers activation to induce accelerated aging) and chemicals (Rapamycin to block TOR and activate autophagy and Bortezomib to block the proteasome) to address the impact on the three different cell populations that comprise the LG, namely the niche (PSC), the progenitor cell population and the differentiated hemocytes. They also induce expression of these transgenes in other tissues with well-known systemic effects (eg. fat body or muscle) and see an effect on LG homeostasis. Authors identify a role of the integrated stress response (ISR) in this process, as genetic or chemical manipulation of ISR was able to partially rescue the effects of accelerated aging in LG homeostasis. I found important concerns on the initial approach to induce aging, on the different results depending on the transgene that is being used, and on the baroque style of writing.

We thank the reviewer for raising this concern on the use of approaches to induce aging and on the different results obtained depending on the transgene being used. In order to address the concern of the reviewer, we have rewritten the entire manuscript and have now focused on the effects caused by the individual transgenes that are being used. We have gotten rid of calling these genetic approaches for either accelerating or decelerating aging but rather use the word “disruption of cellular homeostasis” or “balancing/reinstating cellular homeostasis” as there are multiple reports supporting the role of these molecules/transgenes in controlling cellular homeostasis (Salih and Brunet, 2009; Eijkelenboom and Burgering, 2013; Balisteri et al., 2013; Chun and Kim, 2018; Eskelinen, 2019; Perrotta et al., 2020; Ruland J., 2011; Tabata et al., 2023).

Please see the modified title, abstract, results and discussion section that have been rewritten catering to the comments provided.

Major issues:

- (1) Tools to manipulate cellular aging: on one hand, Foxo and Atg8 overexpression is well known to have an impact on cellular homeostasis (eg. proteostasis) but is not a bona fide tool to induce decelerated cellular aging. On the other hand, boosting autophagy or overexpressing Foxo above physiological levels might have unexpected consequences that are not related to cellular aging. Similar conclusions can be drawn by the use of Imd and Toll activation, which are not a bona fide tool to induce accelerated aging, and whose manipulation might have unexpected consequences in the development of the LG per se. The arguments used by the authors in lines 26-33 to use these genetic tools are not convincing, as these tools are used in other cellular populations and the effects are clearly context dependent.

We agree with the reviewer with this point. Thank you for agreeing that these molecules have been shown to play an important role in regulating cellular homeostasis. We do not claim that these are the only molecules or major regulators of aging or cellular aging. We have used these transgenes as representative approaches to modulate cellular aging.

- Although, we agree with the reviewer that we can't label these tools as Bonafide tools to either accelerate or decelerate aging. Keeping this in mind, we have now rewritten the entire manuscript, modified the title, abstract, results section including the discussion and we project it as genetic perturbation of cellular homeostasis as disruption of cellular homeostasis accompanies organismal or cellular aging. There are numerous reports and review articles some of which have been cited above that support the fact that over-activation of NF- κ B signalling pathways disrupts cellular homeostasis whereas Atg8 or Foxo over-expression maintains cellular homeostasis. We have elaborated and discussed all the results with description of phenotypes obtained as a result of specific transgenes employed. We also agree with the reviewer that we do not see completely opposite or black and white phenotypes, we find that there is some heterogeneity in phenotypes and we mention these heterogenous results obtained and also speculate why these phenotypes might be seen. We agree these are caveats of the study, we are fully aware of this and mention this in our discussion as well. We agree that the transgenes could be playing a context dependent role as cellular aging like the reviewer points out is multifactorial hence the complexity. However, we wanted to investigate the effect of genetic perturbation of these molecules that play an important role in organismal aging, at the cellular level or systemically and what impact this perturbation would have on overall hematopoiesis. We have now toned down our writing, gotten rid of any exaggeration or overstatements and do not call it genetic tools to modulate aging but just cellular homeostasis since cellular homeostasis is indeed perturbed if Toll or Imd are over-activated or Atg8 or Foxo is over-expressed. We also mention possible caveats in the study and our work and discuss possible explanations for the observations seen. We hope that the modified focus and the rewriting of the manuscript will help in addressing the concern of the reviewer. If there are any additional concerns or experimental suggestions that the reviewer has that would strengthen the manuscript, we would be happy to address them.
- (2) In many experiments, the two independent tools to induce decelerated or accelerated aging gave different results on LG homeostasis pointing to gene specific effects and independent of the process of cellular aging itself. Similarly, the impact of Rapamycin also appears to be tool specific.

- We would like to thank the reviewer for pointing this out. Yes, we did observe some transgene specific effects independent of the process of cellular aging itself especially for Foxo overexpression with some Gal4's. We have amply discussed these results and have provided appropriate speculation of why the phenotypes might be caused. In the revised version, we have now gotten rid of calling this as process of cellular aging but rather call it disruption of cellular homeostasis by Toll or Imd activation and balancing/reinstatement of cellular homeostasis by Atg8 or Foxo over-expression. We also agree that Rapamycin shows a rescue of differentiation phenotype for one of the NF κ B pathway activation whereas not the other. This could be because of genetic interaction with one of the pathways and not the other. In this case Imd activation can be rescued by Rapamycin treatment whereas Toll pathway phenotypes cannot be rescued fully, indicating that the inhibition of mTOR pathway by Rapamycin can rescue Imd activation phenotypes whereas Toll cannot be rescued which needs to be probed further using genetic epistasis experiments to understand their genetic interactions.
- (3) Development of the LG takes place in 4 days whereas hematopoiesis in humans lasts the whole life. Whether the fly LG is a bona fide model system to address aging is an overstatement.

LG development starts from the embryonic stages and is fully developed by third instar larval stage and breaks open and releases hemocytes into the circulation during pupariation. We agree the time window of LG development cannot be compared to hematopoiesis in humans that lasts whole life. Here, we are just trying to address the

effect of modulating cellular aging/homeostasis on overall process of hematopoiesis. One of the future goals of this work is to understand what impact does modulation of cellular aging have on blood cells in the adult flies and if differentiated hemocytes that emerge from such progenitors are capable of mounting an effective immune response or not. But we agree that the LG system itself cannot be used to address aging and hence we have removed such statements from the revised version of the manuscript. We have also removed the paragraph on functional implications in understanding ARCH (Age related clonal hematopoiesis) from the manuscript.

Other issues: (1) P62 levels to monitor autophagy flux is not correct. Authors should use atg8 reporters with GFP and RFP, instead.

We would like to thank the reviewer for this comment. We have used the Atg8 antibody instead along with p62 antibody to investigate the autophagy levels. For example: In figure 7 we show that Imd over-activation results in accumulation of p62 and decrease in Atg8 positive puncta whereas ISR up-regulation in this background can rescue abrogation of autophagy as p62 positive puncta go down whereas Atg8 positive puncta increase in number upon genetic rescue. The primary reason for using an antibody instead of the GFP-RFP reporter of Atg8 is because the Gal4 used for genetic modulation in progenitors is recombined with GFP and hence due to technical limitation we could not employ this reporter and have used the antibody instead. We have also changed the word flux to autophagy levels instead.

(2) The Introduction and Discussion sections are extremely long.

Thank you for the comment. We have now cut down on the length of both the sections.

Reviewer 2: Good paper. You switch from ageing to aging. I think it would be uniform to pick one and stick with it throughout the paper.

Thank you so much for finding our paper a good fit for the journal. We have now addressed your suggestion and have switched ageing to aging throughout the manuscript.

Author response to reviewers' comments

Response to reviewers (Version 2 - Minor revision):

I have now reached a decision on the above manuscript.

The reviewer reports are shown at the bottom of this email or can be accessed, together with a copy of this decision letter, by going to:

As you will see, the reviewers are satisfied with the way the manuscript has been reframed and how the comments from the original reviewer were addressed. There are, however, a few quite minor amendments that need to be made. Specifically, 1) as supplemental figures, single channel images need to be provided for Figure 1 panels A-C to visualize p62 and J-N to visualize proteostat, and all of Figure 2 to visualize Antp, 2) a supplemental table of the genetic models used and what they were used for, and 3) more details about imaging (see comments from Reviewer 2). I hope that you will be able to carry these out, because we would like to be able to accept your paper.

We would like to firstly thank the handling editor for reassessing our manuscript and sending it out for review. We are happy we could address the reviewer's comments in the revised version. For the minor amendments suggested above:

- I) We now include a Supplementary Figure 5 having the single channel images to better visualize Figure 1 Panels A-C (for p62) and J-N (for Proteostat) and Supplementary Figure 6 having the single channel images to visualize Antp from Figure 2. These 2 new Supplementary Figures have been included in this revised version of the manuscript.
- II) We have included a table outlining the genetic models used in the study and the purpose they were used for
- III) We have also included the needed information asked by Reviewer 2 on the acquisition parameters for the imaging

All the changes that have been made in the main manuscript have been tracked and highlighted with yellow in the word document

Reviewer 1: In this manuscript, Rohan Jayant Khadilkar and colleagues use *Drosophila* as model system to analyze the potential impact of genetic perturbation of cellular homeostasis on the major hematopoietic organ, the lymph gland (LG). For this purpose, they use a combination of genetic tools (Foxo and Atg8 upregulation to balance cellular homeostasis, and Toll and Imd activation to disrupt cellular homeostasis) and chemicals (Rapamycin to block TOR and activate autophagy and Bortezomib to block the proteasome) to address the impact on the three different cell populations that comprise the LG, namely the niche (PSC), the progenitor cell population and the differentiated hemocytes. They also induce expression of these transgenes in other tissues with well-known systemic effects (eg. fat body or muscle) and see an effect on LG homeostasis. Authors identify a role of the integrated stress response (ISR) in this process, as genetic or chemical manipulation of ISR was able to partially rescue the effects of disrupted cellular homeostasis in LG homeostasis. The paper is well written and present a large collection of data that will be useful to the LG community interested in tissue homeostasis and the role of the ISR in this process.

Minor changes:

- (1) Single channels should be shown in Figures 1 and 2

We would like to thank the reviewer for their constructive comments and feedback that improved the manuscript. Regarding the minor changes suggested - We now include a Supplementary Figure 5 having the single channel images to better visualize Figure 1 Panels A-C (for p62) and J-N (for Proteostat) and Supplementary Figure 6 having the single channel images to visualize Antp from Figure 2. These 2 new Supplementary Figures have been included in this revised version of the manuscript.

Reviewer 2: The manuscript titled "Genetic perturbation of cellular homeostasis regulates Integrated Stress Response signalling to control *Drosophila* hematopoiesis" investigates how disruptions in cellular homeostasis influence hematopoiesis in the *Drosophila* larval lymph gland. The authors employ two opposing genetic strategies: (1) overactivation of NF- κ B signaling pathways (via Toll or Imd), and (2) overexpression of Foxo or Atg8 to counteract NF- κ B activity. These approaches reveal significant changes in the size of the hematopoietic niche, as well as alterations in hemocyte differentiation and overall hematopoietic cell output.

In addition to genetic manipulation, the authors use pharmacological agents—Rapamycin and Bortezomib (a UPS inhibitor)—to modulate hematopoiesis. Mechanistically, the study links these interventions to the Integrated Stress Response (ISR) pathway. Notably, activation of the ISR pathway in the context of Toll or Imd overactivation appears to restore hematopoietic balance, highlighting a potential compensatory mechanism.

By leveraging the *Drosophila* larval lymph gland as a model system, the authors provide valuable insights into how aging-related cellular stress impacts hematopoiesis. The revised manuscript reflects substantial improvements, including a restructured title, abstract, results, and discussion. The central theme—genetic perturbation of cellular homeostasis as a proxy for aging-related stress—is well-articulated.

Figures and Data Presentation:

The figures are well-designed and clearly presented. The revised text is generally easier to follow, although the complexity of the genetic models may still pose challenges for some readers.

Quantitative data are well-documented, with individual data points and variability across experimental conditions clearly shown. The schematic model in Figure 7U is particularly helpful; however, enhancing its resolution (e.g., increasing DPI) would improve its visual clarity.

We would firstly like to thank the reviewer for their critiques that improved our manuscript. We are glad that the reviewer found our manuscript well-articulated and our figures well designed and well presented. We have now addressed the reviewer's comment and uploaded a higher resolution version of Figure 7U and have increased the DPI for improved clarity

Minor Comment:

It would be beneficial to include more detailed information about the objectives used for confocal imaging with the SP8 microscope, to aid reproducibility and technical understanding.

We have included the information on the objectives used for image acquisition along with other necessary parameters in the methods section as suggested by the reviewer for better understanding and reproducibility.

First decision letter

MS ID#: bio.062046

MS Title: Genetic perturbation of cellular homeostasis regulates Integrated Stress Response signalling to control Drosophila hematopoiesis

Authors: Rohan Khadilkar; Kishalay Ghosh; Rohit Krishnan Iyer; Saloni Sood; Mohamed Sabeelil Islam; Jyotsana Labad

Dear Dr Khadilkar,

I have now reached a decision on the above manuscript.

The reviewer reports are shown at the bottom of this email or can be accessed, together with a copy of this decision letter, by going to:

As you will see, the reviewers are satisfied with the way the manuscript has been reframed and how the comments from the original reviewer were addressed. There are, however, a few quite minor amendments that need to be made. Specifically, 1) as supplemental figures, single channel images need to be provided for Figure 1 panels A-C to visualize p62 and J-N to visualize protestat, and all of Figure 2 to visualize Antp, 2) a supplemental table of the genetic models used and what they were used for, and 3) more details about imaging (see comments from Reviewer 2). I hope that you will be able to carry these out, because we would like to be able to accept your paper.

Comments from the Reviewers:

In this manuscript, Rohan Jayant Khadilkar and colleagues use Drosophila as model system to analyze the potential impact of genetic perturbation of cellular homeostasis on the major hematopoietic organ, the lymph gland (LG). For this purpose, they use a combination of genetic tools (Foxo and Atg8 upregulation to balance cellular homeostasis, and Toll and Imd activation to disrupt cellular homeostasis) and chemicals (Rapamycin to block TOR and activate autophagy and Bortezomib to block the proteasome) to address the impact on the three different cell populations that comprise the LG, namely the niche (PSC), the progenitor cell population and the differentiated hemocytes. They also induce expression of these transgenes in other tissues with well-known systemic effects (eg. fat body or muscle) and see an effect on LG homeostasis. Authors identify a role of the integrated stress response (ISR) in this process, as genetic or chemical manipulation of ISR was able to partially rescue the effects of disrupted cellular homeostasis in LG homeostasis. The

paper is well written and present a large collection of data that will be useful to the LG community interested in tissue homeostasis and the role of the ISR in this process.

Minor changes:

(1) Single channels should be shown in Figures 1 and 2

Reviewer 2: The manuscript titled "Genetic perturbation of cellular homeostasis regulates Integrated Stress Response signalling to control *Drosophila* hematopoiesis" investigates how disruptions in cellular homeostasis influence hematopoiesis in the *Drosophila* larval lymph gland. The authors employ two opposing genetic strategies: (1) overactivation of NF- κ B signaling pathways (via Toll or Imd), and (2) overexpression of Foxo or Atg8 to counteract NF- κ B activity. These approaches reveal significant changes in the size of the hematopoietic niche, as well as alterations in hemocyte differentiation and overall hematopoietic cell output.

In addition to genetic manipulation, the authors use pharmacological agents—Rapamycin and Bortezomib (a UPS inhibitor)—to modulate hematopoiesis. Mechanistically, the study links these interventions to the Integrated Stress Response (ISR) pathway. Notably, activation of the ISR pathway in the context of Toll or Imd overactivation appears to restore hematopoietic balance, highlighting a potential compensatory mechanism.

By leveraging the *Drosophila* larval lymph gland as a model system, the authors provide valuable insights into how aging-related cellular stress impacts hematopoiesis. The revised manuscript reflects substantial improvements, including a restructured title, abstract, results, and discussion. The central theme—genetic perturbation of cellular homeostasis as a proxy for aging-related stress—is well-articulated.

Figures and Data Presentation:

The figures are well-designed and clearly presented. The revised text is generally easier to follow, although the complexity of the genetic models may still pose challenges for some readers. Quantitative data are well-documented, with individual data points and variability across experimental conditions clearly shown. The schematic model in Figure 7U is particularly helpful; however, enhancing its resolution (e.g., increasing DPI) would improve its visual clarity.

Minor Comment:

It would be beneficial to include more detailed information about the objectives used for confocal imaging with the SP8 microscope, to aid reproducibility and technical understanding.

Reviewer's Responses to Questions

Experimental quality

Does each figure have the proper controls?

If 'No', please indicate reasons in Comments for Author box below.

Reviewer #1:

Yes

Reviewer #2:

Yes

Were the data analyzed using appropriate statistical tests?

If 'No', please indicate reasons in Comments for Author box below.

Reviewer #1:

Yes

Reviewer #2:

Yes

Reproducibility

Were experiments performed using adequate number of biological replicates?
If 'No', please indicate reasons in Comments for Author box below.

Reviewer #1:

Yes

Reviewer #2:

Yes

Does the methods section provide sufficient detail to permit reproducibility?
If 'No', please indicate reasons in Comments for Author box below.

Reviewer #1:

Yes

Reviewer #2:

Yes

Completeness

Are the manuscript's conclusions supported by the data?
If 'No', please indicate reasons in Comments for Author box below.

Reviewer #1:

Yes

Reviewer #2:

Yes

Scholarship

Do the authors cite and discuss the merits of data that would argue for and against their conclusion?
If 'No', please indicate reasons in Comments for Author box below.

Reviewer #1:

Yes

Reviewer #2:

Yes

Does the manuscript title & abstract accurately reflect the contents of the manuscript, without hyperbole?

If 'No', please indicate reasons in Comments for Author box below.

Reviewer #1:

Yes

Reviewer #2:

Yes

First revision

Author response to reviewers' comments

Reviewer 1

In this manuscript, Rohan Jayant Khadilkar and colleagues use *Drosophila* as model system to analyze the potential impact of aging on the homeostasis of the major hematopoietic organ, the lymph gland (LG). For this purpose, they use a combination of genetic tools (Foxo and Atg8 upregulation to induce decelerated aging, and Toll and Imd Response To Reviewers activation to induce accelerated aging) and chemicals (Rapamycin to block TOR and activate autophagy and Bortezomib to block the proteasome) to address the impact on the three different cell populations that comprise the LG, namely the niche (PSC), the progenitor cell population and the differentiated hemocytes. They also induce expression of these transgenes in other tissues with well-known systemic effects (eg. fat body or muscle) and see an effect on LG homeostasis. Authors identify a role of the integrated stress response (ISR) in this process, as genetic or chemical manipulation of ISR was able to partially rescue the effects of accelerated aging in LG homeostasis. I found important concerns on the initial approach to induce aging, on the different results depending on the transgene that is being used, and on the baroque style of writing.

We thank the reviewer for raising this concern on the use of approaches to induce aging and on the different results obtained depending on the transgene being used. In order to address the concern of the reviewer, we have rewritten the entire manuscript and have now focused on the effects caused by the individual transgenes that are being used. We have gotten rid of calling these genetic approaches for either accelerating or decelerating aging but rather use the word “disruption of cellular homeostasis” or “balancing/reinstating cellular homeostasis” as there are multiple reports supporting the role of these molecules/transgenes in controlling cellular homeostasis (Salih and Brunet, 2009; Eijkelenboom and Burgering, 2013; Balisteri et al., 2013; Chun and Kim, 2018; Eskelinen, 2019; Perrotta et al., 2020; Ruland J., 2011; Tabata et al., 2023).

Please see the modified title, abstract, results and discussion section that have been rewritten catering to the comments provided.

Major issues:

- (4) Tools to manipulate cellular aging: on one hand, Foxo and Atg8 overexpression is well known to have an impact on cellular homeostasis (eg. proteostasis) but is not a bona fide tool to induce

decelerated cellular aging. On the other hand, boosting autophagy or overexpressing Foxo above physiological levels might have unexpected consequences that are not related to cellular aging. Similar conclusions can be drawn by the use of Imd and Toll activation, which are not a bona fide tool to induce accelerated aging, and whose manipulation might have unexpected consequences in the development of the LG per se. The arguments used by the authors in lines 26-33 to use these genetic tools are not convincing, as these tools are used in other cellular populations and the effects are clearly context dependent.

We agree with the reviewer with this point. Thank you for agreeing that these molecules have been shown to play an important role in regulating cellular homeostasis. We do not claim that these are the only molecules or major regulators of aging or cellular aging. We have used these transgenes as representative approaches to modulate cellular aging. Although, we agree with the reviewer that we can't label these tools as Bonafide tools to either accelerate or decelerate aging. Keeping this in mind, we have now rewritten the entire manuscript, modified the title, abstract, results section including the discussion and we project it as genetic perturbation of cellular homeostasis as disruption of cellular homeostasis accompanies organismal or cellular aging. There are numerous reports and review articles some of which have been cited above that support the fact that over-activation of NF- κ B signalling pathways disrupts cellular homeostasis whereas Atg8 or Foxo over-expression maintains cellular homeostasis. We have elaborated and discussed all the results with description of phenotypes obtained as a result of specific transgenes employed. We also agree with the reviewer that we do not see completely opposite or black and white phenotypes, we find that there is some heterogeneity in phenotypes and we mention these heterogenous results obtained and also speculate why these phenotypes might be seen. We agree these are caveats of the study, we are fully aware of this and mention this in our discussion as well. We agree that the transgenes could be playing a context dependent role as cellular aging like the reviewer points out is multifactorial hence the complexity. However, we wanted to investigate the effect of genetic perturbation of these molecules that play an important role in organismal aging, at the cellular level or systemically and what impact this perturbation would have on overall hematopoiesis. We have now toned down our writing, gotten rid of any exaggeration or overstatements and do not call it genetic tools to modulate aging but just cellular homeostasis since cellular homeostasis is indeed perturbed if Toll or Imd are over-activated or Atg8 or Foxo is over-expressed. We also mention possible caveats in the study and our work and discuss possible explanations for the observations seen. We hope that the modified focus and the rewriting of the manuscript will help in addressing the concern of the reviewer. If there are any additional concerns or experimental suggestions that the reviewer has that would strengthen the manuscript, we would be happy to address them.

- (5) In many experiments, the two independent tools to induce decelerated or accelerated aging gave different results on LG homeostasis pointing to gene specific effects and independent of the process of cellular aging itself. Similarly, the impact of Rapamycin also appears to be tool specific.

We would like to thank the reviewer for pointing this out. Yes, we did observe some transgene specific effects independent of the process of cellular aging itself especially for Foxo overexpression with some Gal4's. We have amply discussed these results and have provided appropriate speculation of why the phenotypes might be caused. In the revised version, we have now gotten rid of calling this as process of cellular aging but rather call it disruption of cellular homeostasis by Toll or Imd activation and balancing/reinstatement of cellular homeostasis by Atg8 or Foxo over-expression. We also agree that Rapamycin shows a rescue of differentiation phenotype for one of the NF κ B pathway activation whereas not the other. This could be because of genetic interaction with one of the pathways and not

the other. In this case Imd activation can be rescued by Rapamycin treatment whereas Toll pathway phenotypes cannot be rescued fully, indicating that the inhibition of mTOR pathway by Rapamycin can rescue Imd activation phenotypes whereas Toll cannot be rescued which needs to be probed further using genetic epistasis experiments to understand their genetic interactions.

- (6) Development of the LG takes place in 4 days whereas hematopoiesis in humans lasts the whole life. Whether the fly LG is a bona fide model system to address aging is an overstatement.

LG development starts from the embryonic stages and is fully developed by third instar larval stage and breaks open and releases hemocytes into the circulation during pupariation. We agree the time window of LG development cannot be compared to hematopoiesis in humans that lasts whole life. Here, we are just trying to address the effect of modulating cellular aging/homeostasis on overall process of hematopoiesis. One of the future goals of this work is to understand what impact does modulation of cellular aging have on blood cells in the adult flies and if differentiated hemocytes that emerge from such progenitors are capable of mounting an effective immune response or not. But we agree that the LG system itself cannot be used to address aging and hence we have removed such statements from the revised version of the manuscript. We have also removed the paragraph on functional implications in understanding ARCH (Age related clonal hematopoiesis) from the manuscript.

Other issues: (1) P62 levels to monitor autophagy flux is not correct. Authors should use atg8 reporters with GFP and RFP, instead.

We would like to thank the reviewer for this comment. We have used the Atg8 antibody instead along with p62 antibody to investigate the autophagy levels. For example: In figure 7 we show that Imd over-activation results in accumulation of p62 and decrease in Atg8 positive puncta whereas ISR up-regulation in this background can rescue abrogation of autophagy as p62 positive puncta go down whereas Atg8 positive puncta increase in number upon genetic rescue. The primary reason for using an antibody instead of the GFP-RFP reporter of Atg8 is because the Gal4 used for genetic modulation in progenitors is recombined with GFP and hence due to technical limitation we could not employ this reporter and have used the antibody instead. We have also changed the word flux to autophagy levels instead.

- (2) The Introduction and Discussion sections are extremely long.

Thank you for the comment. We have now cut down on the length of both the sections.

Reviewer 2: Good paper. You switch from ageing to aging. I think it would be uniform to pick one and stick with it throughout the paper.

Thank you so much for finding our paper a good fit for the journal. We have now addressed your suggestion and have switched ageing to aging throughout the manuscript.

Second decision letter

MS ID#: bio.062046R1

MS Title: Genetic perturbation of cellular homeostasis regulates Integrated Stress Response signalling to control Drosophila hematopoiesis

Authors: Rohan Khadilkar; Kishalay Ghosh; Rohit Krishnan Iyer; Saloni Sood; Mohamed Sabeelil Islam; Jyotsana Labad

Dear Dr Khadilkar,

Thank you for adding new supplemental figures, providing a table of the lines used, and incorporating more details about image acquisition. I am happy to tell you that your manuscript has been accepted for publication in Biology Open, pending our standard publication integrity checks. It was accepted on 5 June 2025